# Developmental hourglass and heterochronic shifts in fin and limb development

**Koh Onimaru[1,2,3]\*, Kaori Tatsumi[1], Chiharu Tanegashima[1], Mitsutaka Kadota[1], Osamu Nishimura[1], Shigehiro Kuraku[1]\***

[1]Laboratory for Phyloinformatics, RIKEN Center for Biosystems Dynamics Research (BDR), Kobe, Japan; [2]Laboratory for Bioinformatics Research, RIKEN BDR, Wako City, Japan; [3]Molecular Oncology Laboratory, Graduate School of Medicine, Nagoya University, Nagoya, Japan

**Abstract** How genetic changes are linked to morphological novelties and developmental constraints remains elusive. Here, we investigate genetic apparatuses that distinguish fish fins from tetrapod limbs by analyzing transcriptomes and open-chromatin regions (OCRs). Specifically, we compared mouse forelimb buds with the pectoral fin buds of an elasmobranch, the brown-banded bamboo shark (*Chiloscyllium punctatum*). A transcriptomic comparison with an accurate orthology map revealed both a mass heterochrony and hourglass-shaped conservation of gene expression between fins and limbs. Furthermore, open-chromatin analysis suggested that access to conserved regulatory sequences is transiently increased during mid-stage limb development. During this stage, stage-specific and tissue-specific OCRs were also enriched. Together, early and late stages of fin/limb development are more permissive to mutations than middle stages, which may have contributed to major morphological changes during the fin-to-limb evolution. We hypothesize that the middle stages are constrained by regulatory complexity that results from dynamic and tissue-specific transcriptional controls.

**\*For correspondence:**
koh.onimaru@riken.jp (KO);
shigehiro.kuraku@riken.jp (SK)

**Competing interests:** The authors declare that no competing interests exist.

## Introduction

The genetic mechanism of morphological diversity among multicellular organisms is of central interest in evolutionary biology. In particular, our understanding of how morphological novelties are linked to the emergence of their respective genetic apparatuses is limited (*Rebeiz and Tsiantis, 2017*). In addition, it is still unclear to what extent internal constraints, such as pleiotropy, affect evolvability (*Wagner and Zhang, 2011*). The fin-to-limb transition is a classic, yet still influential, case study that contributes to our understanding of morphological evolution. In general, tetrapod limbs are composed of three modules, the stylopod, zeugopod, and autopod, which are ordered proximally to distally (*Figure 1A*). In contrast, fish fins are often subdivided into different anatomical modules along the anterior−posterior axis—the propterygium, mesopterygium, and metapterygium (*Figure 1A*). Although it is still controversial how this different skeletal arrangement compares with the archetypal tetrapod limb, the autopod (wrist and digits) seems to be the most apparent morphological novelty during the fin-to-limb transition (*Clack, 2009*). Despite intensive comparative studies of developmental gene regulation, genetic machinery that differs between fins and limbs remains elusive. Instead, several studies revealed that autopod-specific regulation of *Hoxa13* and *Hoxd10−13*, which control autopod formation, is also conserved in non-tetrapod vertebrates (*Davis et al., 2007*; *Freitas et al., 2007*; *Schneider et al., 2011*), except that the expression domains of *Hoxa13* and *Hoxa11* are mutually exclusive in mouse and chick limbs while overlapping in examined fish fin buds (note that axolotl limbs also exhibit such fish-like overlap of these

**eLife digest** Animals come in all shapes and sizes. This diversity arose through genetic mutations during evolution, but it is unclear exactly how these variations led to the formation of new shapes. There is increasing evidence to suggest that not all shapes are possible and that variability between animals is limited by a phenomenon known as "developmental constraint". These limitations direct parts of the body towards a specific shape as they develop in the embryo. Therefore, understanding the mechanisms underlying these developmental constraints could help explain how different body shapes evolved.

The limbs of humans and other mammals evolved from the fins of fish, and this transition is often used to study the role developmental constraints play in evolution. This is an ideal model as there is already a detailed fossil record mapping this evolutionary event, and data pinpointing some of the genes involved in the development of limbs and fins. But this data is incomplete, and a full comparison between the genes activated in the fin and the limb during embryonic development had not been achieved. This is because most fish used for research have undergone recent genetic changes, making it hard to spot which genetic differences are linked to the evolution of the limb.

To overcome this barrier, Onimaru et al. compared genetic data from the developing limbs of mice to the developing fins of the brown-banded bamboo shark, which evolves much slower than other fish. This revealed that although many genes commonly played a role in the development of the fin and the limb in the embryo, the activity of these shared genes was not the same. For example, genes that switched on in the late stages of limb development, switched off in the late stages of fin development. But in the middle of development, those differences were relatively small and both species activated very similar sets of genes. Many of these genes were pleiotropic, which means they have important roles in other tissues and therefore mutate less often. This suggests that the mid-stage of limb development is under the strongest level of constraint.

Darwin's theory of natural selection explains that mutations drive evolution. But the theory cannot predict what kinds of new body shapes new mutations will produce. Understanding how the activity levels of different genes affect development could help to fill this knowledge gap. This has potential medical applications, for example, understanding why some genetic changes cause more serious problems than others. This work suggests that mutations in genes that are active during the mid-stage of limb development may have the most serious impact.

expression domains; *Ahn and Ho, 2008*; *Metscher et al., 2005*; *Sakamoto et al., 2009*; *Woltering et al., 2019*). Although several gene regulatory differences have been proposed to explain the anatomical difference between fins and limbs, these proposals have been exclusively focused on *Hox* genes (*Kherdjemil et al., 2016*; *Nakamura et al., 2016*; *Sheth et al., 2012*; *Woltering et al., 2014*). Therefore, a genome-wide systematic study is required to identify the genetic differences between fish fins and tetrapod limbs.

There have been several difficulties that limit genetic comparisons between tetrapods and non-tetrapod vertebrates. For example, whereas zebrafish and medaka are ideal models for molecular studies, their rapid evolutionary speed and a teleost-specific whole-genome duplication hinder comparative analyses with tetrapods at both the morphological and genetic levels (*Ravi and Venkatesh, 2008*). This obstacle can be circumvented by using more slowly evolving species such as spotted gar, coelacanths, and elephantfish (also known as elephant shark, a cartilaginous fish that is not a true shark) with their genome sequences that have not experienced recent lineage-specific genome duplications and thus facilitate the tracing of the evolution of gene regulation (*Amemiya et al., 2013*; *Braasch et al., 2016*). However, the major disadvantage of these slowly evolving species is the inaccessibility of developing embryos. In contrast, although the eggs of sharks and rays (other slowly evolving species; *Hara et al., 2018*) are often more accessible, their genomic sequence information has not been available until recently. As a solution for these problems, this study used embryos of the brownbanded bamboo shark (referred to hereafter as the bamboo shark), because a usable genome assembly was recently published for this species (*Hara et al., 2018*). Importantly, its non-coding sequences seem to be more comparable with those of tetrapods than with teleosts (*Hara et al., 2018*). In addition, this species is common in aquariums and has a detailed

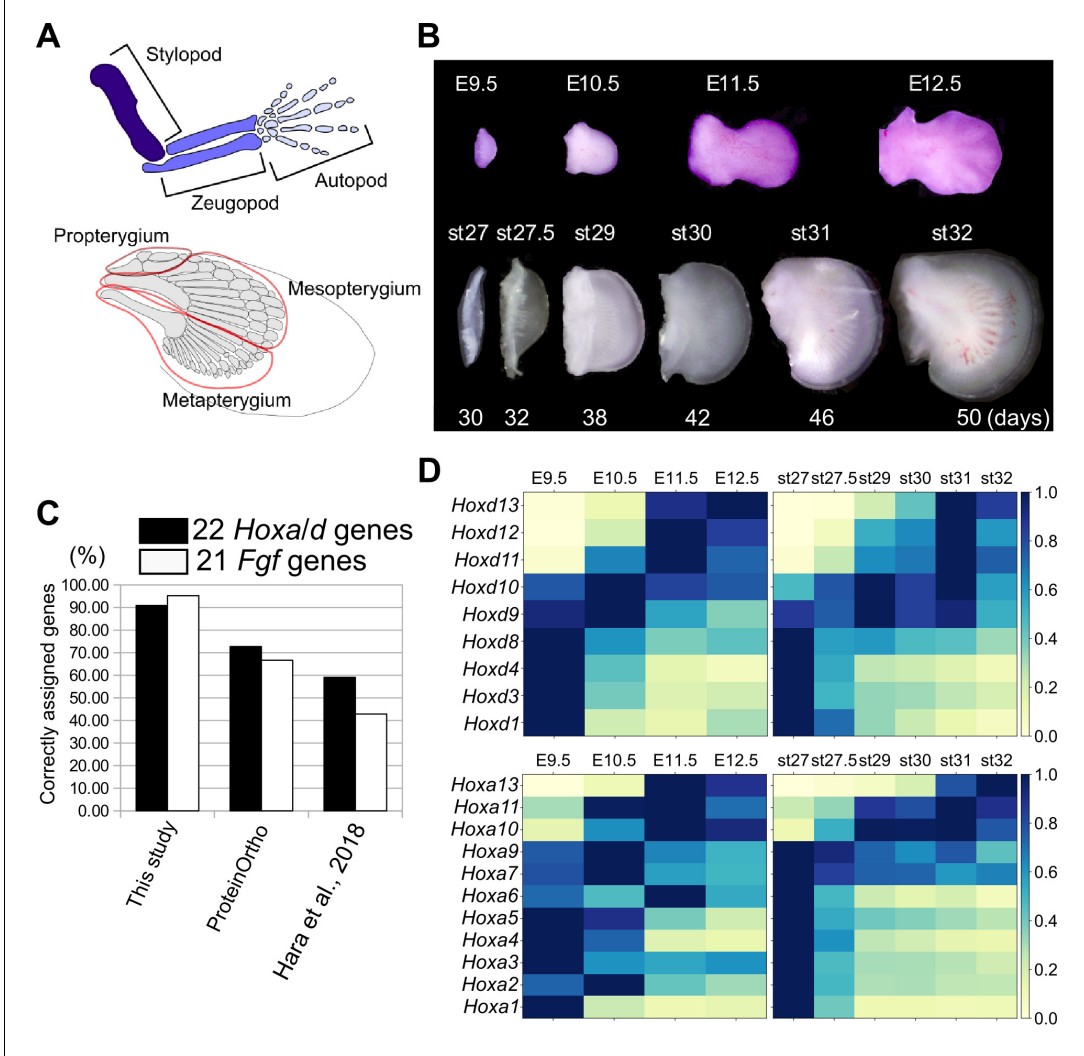

**Figure 1.** Transcriptome analysis and orthology assignment. (**A**) The skeletal patterns of a mouse limb (top) and a bamboo shark pectoral fin (bottom). Anterior is to the top; distal is to the right. (**B**) Mouse forelimb buds and bamboo shark pectoral fin buds that were analyzed by RNA-seq. (**C**) Comparison of the accuracy of three orthology assignment methods. Vertical axis, the percentages of correctly assigned *Hoxa* and *Hoxd* paralogs (black bars) and *Fgf* paralogs (white bars). (**D**) Heat map visualization of the transcription profile of *Hoxa* and *Hoxd* genes in mouse limb buds (left) and bamboo shark fin buds (right) with scaled TPMs.

The online version of this article includes the following figure supplement(s) for figure 1:

**Figure supplement 1.** Schematic representation of the orthology assignment algorithm.
**Figure supplement 2.** Molecular phylogenetic tree for Fgf family.
**Figure supplement 3.** Additional molecular phylogenetic trees for Fgf8, Fgf11, Fgf12, and Fgf13.
**Figure supplement 4.** Comparison between the TPM and TMM.
**Figure supplement 5.** The effect of scaling methods to housekeeping genes.
**Figure supplement 6.** The effect of scaling methods to heterochronic genes.
**Figure supplement 7.** Examination of quantitative collinearity of *Hoxd* genes.
**Figure supplement 8.** Expression profile of genes related to cellular differentiation.

developmental staging table, providing an opportunity to study embryogenesis (*Onimaru et al., 2018*). These unique circumstances of the bamboo shark enabled a comprehensive study to identify the genetic differences between fins and limbs.

In this study, to identify genetic differences between fins and limbs, we performed RNA sequencing (RNA-seq) analyses of developing bamboo shark fins and mouse limbs. Along with this transcriptomic comparison, we also generated an accurate orthology map between the bamboo shark and mouse. In addition, we applied an assay for transposase-accessible chromatin with high-throughput and chromatin accessibility analysis (ATAC-seq; *Buenrostro et al., 2013*) across a time series of mouse limb buds, which generated a high-quality data set the showing dynamics of open-chromatin regions (OCRs; putative enhancers) during limb development. We also analyzed the evolutionary conservation of sequences in these OCRs to gain insights into the gene regulatory changes during the fin-to-limb transition.

## Results

### Comparative transcriptome analysis

To compare the temporal dynamics of gene expression between bamboo shark fin and mouse limb development, we obtained RNA-seq data from a time series of growing fin and limb buds with three replicates (*Figure 1B*; *Supplementary file 1* for the details of RNA-seq). We selected limb buds from embryonic day (E)9.5 to E12.5 mice because this is the period during which the major segments of the tetrapod limb—the stylopod, zeugopod, and autopod—become apparent. In particular, the presumptive autopod domain, which is a distinct structure in the tetrapod limb, is visually recognizable from E11.5. For the bamboo shark, we selected developing fin stages from as wide a time period as possible (*Figure 1B*). To perform fine-scale molecular-level comparison, we annotated its coding genes using BLASTP against several vertebrates (listed in the Materials and methods) and our custom algorithm. As a result, 16443 unique genes from 63898 redundant coding transcripts were annotated as orthologous to known genes of vertebrates, among which 13,005 genes were uniquely orthologous to mouse genes (*Table 1* for details of the transcriptome assembly; *Figure 1—figure supplement 1—3*, *Supplementary files 2* and *3* for gene annotations and Supplementary data for sequence information). The number of detected orthologs is reasonable when compared with other studies (e.g. *Hao et al., 2020*). The quality of the ortholog assignment, which was assessed by examining *Hox* and *Fgf* genes, showed that our custom algorithm is more accurate than other methods (*Figure 1C*; see Materials and Methods and *Supplementary file 4* for details). Using this assembly for the bamboo shark and RefSeq genes for mice, the means and standard errors of the transcripts per million (TPM) values were calculated from three replicates (see *Figure 1—figure supplement 4* for other normalization methods and *Supplementary files 5* and *6* for the full list of TPM values). In addition, for most of the analyses, TPMs were scaled by setting the highest TPM in

**Table 1.** Assembly statistics of bamboo shark transcriptome.

| Characteristic | Bamboo shark transcriptome | Bamboo shark gene model (*Hara et al., 2018*) |
|---|---|---|
| Total number of sequences | 222015 | 34038 |
| Total sequence length (bp) | 195541367 | 36633751 |
| Average length (bp) | 880 | 1076 |
| Maximum length (bp) | 18451 | 108594 |
| N count | 0 | 10208 |
| L50 | 24765 | 5666 |
| N50 length (bp) | 2075 | 1749 |
| Protein coding | 63898 | 34038 |
| Orthology detected | 41633 | 18180 |
| Unique orthologs | 14139 | 14907 |
| Unique orthologs without gene symbols | 1821 | 1780 |
| Unique orthologs only in elephantfish | 826 | 552 |
| Sequences with no orthology | 20892 | 15254 |
| Orthologs with mouse genes | 12326 | 13005 |

each gene of each species to '1' (which we refer to as the Max one method) to capture temporal dynamics rather than absolute transcript amounts. Compared to using intact TPMs and other scaling methods, Max one is relatively sensitive to interspecific differences in dynamically regulated gene expression (see Materials and methods and *Figure 1—figure supplements 5* and *6* for details).

With this transcriptome data set and gene annotation, we first validated our data by analyzing the expression profiles of *Hoxa* and *Hoxd* genes. In mouse limb development, *Hoxa* and *Hoxd* genes undergo two phases of global regulation (*Deschamps and Duboule, 2017*). During the first phase, *Hoxd* genes are regulated by an enhancer group located 3' of the entire HoxD cluster, and the *Hoxd* genes are sequentially upregulated from 3' to 5'. The outcome of this first phase helps to establish the arm and the forearm. During the second phase, enhancers located 5' of the HoxD cluster start to activate expression of *Hoxd10* to *Hoxd13* in the presumptive autopod region (Hoxa genes are regulated in a similar manner; *Deschamps and Duboule, 2017*). As expected, we detected the two phases of *Hoxd* gene regulation in mouse limb transcriptomes; the expression levels of *Hoxd1* to *Hoxd8* were highest at E9.5 (the first phase regulation), and *Hoxd11* to *Hoxd13* were gradually upregulated later (the second phase regulation; *Figure 1D*). Interestingly, the expression levels of *Hoxd9* and *Hoxd10* were highest at E10.5, which probably represents the transitional state between the first and second global regulation (*Andrey et al., 2013*). A similar profile was observed for *Hoxa* genes (*Figure 1D*). As with mouse limb buds, we found similar phasic regulation of *Hoxa* and *Hoxd* genes in the bamboo shark fin transcriptome (*Figure 1D*), suggesting that these transcriptomic data cover comparable developmental stages between the two species at least with respect to *Hox* gene regulation.

The overall similarity in the temporal dynamics of *Hox* gene expression between the mouse limb bud and the bamboo shark fin bud is an expected result because the second phase of *Hoxd* gene regulation has been found to be conserved in the fins of many fish (*Ahn and Ho, 2008*; *Davis et al., 2007*; *Freitas et al., 2007*; *Schneider et al., 2011*; *Tulenko et al., 2017*). However, there are several differences that are worth noting. For example, in mouse limb buds, *Hoxd11* and *Hoxd12* expression was highest at E11.5, followed by further upregulation of *Hoxd13* at E12.5 (*Figure 1D*). In contrast, in bamboo shark fin buds, these three genes reached their peak expression simultaneously at [stage (st)]31 (*Figure 1D*). This led us to investigate further whether the quantitative collinearity of 5' *Hoxd* genes, where the expression of *Hoxd13* is much higher than that of its neighboring *Hoxd* genes, whose transcription levels decrease with increasing distance from *Hoxd13* (*Montavon et al., 2008*), is conserved in the bamboo shark fin buds. First, as a confirmation of the previous observation, we also found quantitative collinearity of *Hoxd* genes in our transcriptome data of mouse limb buds at E12.5 (*Figure 1—figure supplement 7*). However, the bamboo shark fin buds exhibited no clear relationship between the genomic loci and the expression levels of *Hoxd* genes at either st31 or st32 (*Figure 1—figure supplement 7*): *Hoxd12* expression was highest among its neighbors. *Hoxd9* showed the second highest expression, followed by *Hoxd10* and *Hoxd11*, which had roughly identical levels of transcripts. *Hoxd13* expression was lowest among these 5' members. Given that quantitative collinearity is considered to be a consequence of the characteristic global regulation of the HoxD cluster in the mouse limb bud (*Montavon et al., 2008*), this result suggests that the bamboo shark fin bud may have a different mechanism for *Hoxd* gene regulation. Interestingly, a recent study also showed that the presumptive autopod domains of chick limb buds express nearly a same amount of *Hoxd13* and *Hoxd12* transcripts (*Yakushiji-Kaminatsui et al., 2018*), suggesting that quantitative collinearity is not a universal feature of fins and limbs, rather varies among species. Taken together, although the overall temporal dynamics of *Hox* gene expression are conserved between the mouse limb bud and the bamboo shark fin bud, some differences in the regulation of *Hox* genes may exist between species.

To investigate to what extent our bulk transcriptome data captured the processes of cellular differentiation, we also analyzed genes related to chrondrogenesis and myogenesis. As a result, we found that the chondrogenic pathway was at least partially conserved between bambooshark fin buds and mouse limb buds; the expression level of *Sox9* and *Runx3* (key transcription factors of chondrogenesis; *Fowler and Larsson, 2020*) increased relatively early, and that of *Acan* (a cartilage-specific proteoglycan; *Fowler and Larsson, 2020*) was upregulated later (*Figure 1—figure supplement 8*). In contrast, although *Nog* is known to be expressed in cartilaginous condensations in mouse limb buds (*Brunet et al., 1998*), we did not detect a *Nog* ortholog in either the fin transcriptome or the genome assembly of the bamboo shark. As for myogenesis, our transcriptome data

captured both conserved and divergent myogenetic regulation: *Pax3* (a marker of myogenic precursor cells) was downregulated over developmental time, and the MyoD gene family (*Myog*, *Myod1*, *Myf5*) took turns for further differentiation (*Chal and Pourquié, 2017*). In contrast, whereas mouse limb buds showed upregulation of three myosin genes (*Myh3*, *Myh7*, *Myh8*) at E12.5, we detected the upregulation of only *Myh7* in bamboo shark fin buds. Again, we did not find *Myh3* and *Myh8* in either the transcriptome or the genome assembly of the bamboo shark. These results suggest that our transcriptome data, even though based on bulk sampling of RNA, can reveal conserved and diverged cellular differentiation processes.

## Heterochronic gene expressions

Next, to find differences in gene regulation between the two species, we performed a gene-by-gene comparison of expression dynamics with hierarchical clustering (*Figure 2A*). To find potential candidate genes that contribute to the different morphologies between fins and limbs, we annotated genes with mouse mutant phenotypes (see *Supplementary file 7* for the full list of genes, expression data, and annotation). The result showed that 6701 genes were significantly expressed in only one of these species ('Fin-specific' and 'Limb-specific' in *Figure 2A*; 3284 and 3417 genes, respectively). While the fin-specific gene group consisted of many uncharacterized genes, it included ones that are known to control only fish fin development (*Fischer et al., 2003*; *Zhang et al., 2010*), such as *And1* (TRINITY_DN62789_c1_g1_i3 in Supplementary data; ortholog of a coelacanth gene, XP_015216565) and *Fgf24* (TRINITY_DN92536_c7_g1_i2 in Supplementary data; ortholog of a coelacanth gene, XP_006012032). In the limb-specific gene group, several interesting genes were listed that exhibit abnormal phenotype in the mouse limb (e.g. *Bmp2, Ihh*, and *Megf8*). However, the number of these species-specific genes is probably unreliable and overestimated because these groups also contain genes for which their orthology was not assigned correctly. We also detected 1884 genes that were upregulated during late stages of fin/limb development for both species, including genes that are well known to be expressed later during fin/limb development, such as the autopod-related transcription factors *Hoxd13* and *Hoxa13* and differentiation markers *Col2a1* and *Mef2c* ('Conserved, late1 and Conserved, late2' in *Figure 2A*). Intriguingly, 5388 genes exhibited heterochronic expression profiles; their expression levels were highest during the late stages of mouse limb bud development but were relatively stable expression throughout fin development ('Heterochronic1'; 3178 genes) or decreased during the late stages of fin development ('Heterochronic2'; 2223 genes; see *Supplementary file 7* for the full list of genes and annotations). For validation, we examined the spatio-temporal expression pattern of three heterochronic genes that exhibit limb abnormality in mouse mutants, *Aldh1a2* from Heterochronic1 and, *Hand2* and *Vcan* from Heterochronic2. *Aldh1a2* is upregulated in the interdigital web of mouse limb buds from E11.5 (*Figure 2—figure supplement 1A*) and known to positively regulate interdigital cell death (*Kuss et al., 2009*). On the other hand, in bamboo shark fin buds, *Aldh1a2* expression was initially uniform and was later restricted to the distal edge of fin buds (*Figure 2—figure supplement 1A*). *Hand2* and *Vcan* transcripts were upregulated in mouse forelimb buds at E12.5 and downregulated in bamboo shark fin buds at st32 (*Figure 2B,C*). Thus, the temporal transcriptomic profiles were consistent with spatial expression patterns.

For a comparison, we found relatively few genes that were downregulated over time in the mouse limb bud but were upregulated in the shark fin. There was a total of 241 such genes, but only 43 of them displayed a clear heterochrony (yellow empty box in *Figure 2—figure supplement 1B* and *Supplementary file 8* for the list of the genes). Of those, *Fgf8* is particularly interesting as FGF8 plays a crucial role as a growth signal from the apical ectodermal ridge (AER) in mouse and chick limb buds (*Lewandoski et al., 2000*). As shown in *Figure 2—figure supplement 1C*, *Fgf8* expression was high during the early stages of limb buds and was gradually downregulated at later stages. In contrast, in bamboo shark fin buds, *Fgf8* was expressed very weakly (around 0.1 TPM) at st. 27 and st. 27.5 and was upregulated at later stages. Indeed, this late upregulation of *Fgf8* was also reported in the apical fin fold (roughly equivalent to the AER) of zebrafish pectoral fin buds (*Nomura et al., 2006*). In the zebrafish pectoral fin bud, *Fgf16* and *Fgf24* are upregulated earlier than *Fgf8* (*Draper et al., 2003*; *Nomura et al., 2006*). In addition, *Fgf4*, *Fgf9*, and *Fgf17* are expressed in the AER and have a redundant function in the mouse limb bud (*Mariani et al., 2008*). Therefore, we also examined these other *Fgf* genes and found that moderate expression of *Fgf9*, *Fgf16*, and *Fgf24* were detected in the early stages of bamboo shark fin buds (*Figure 2—figure*

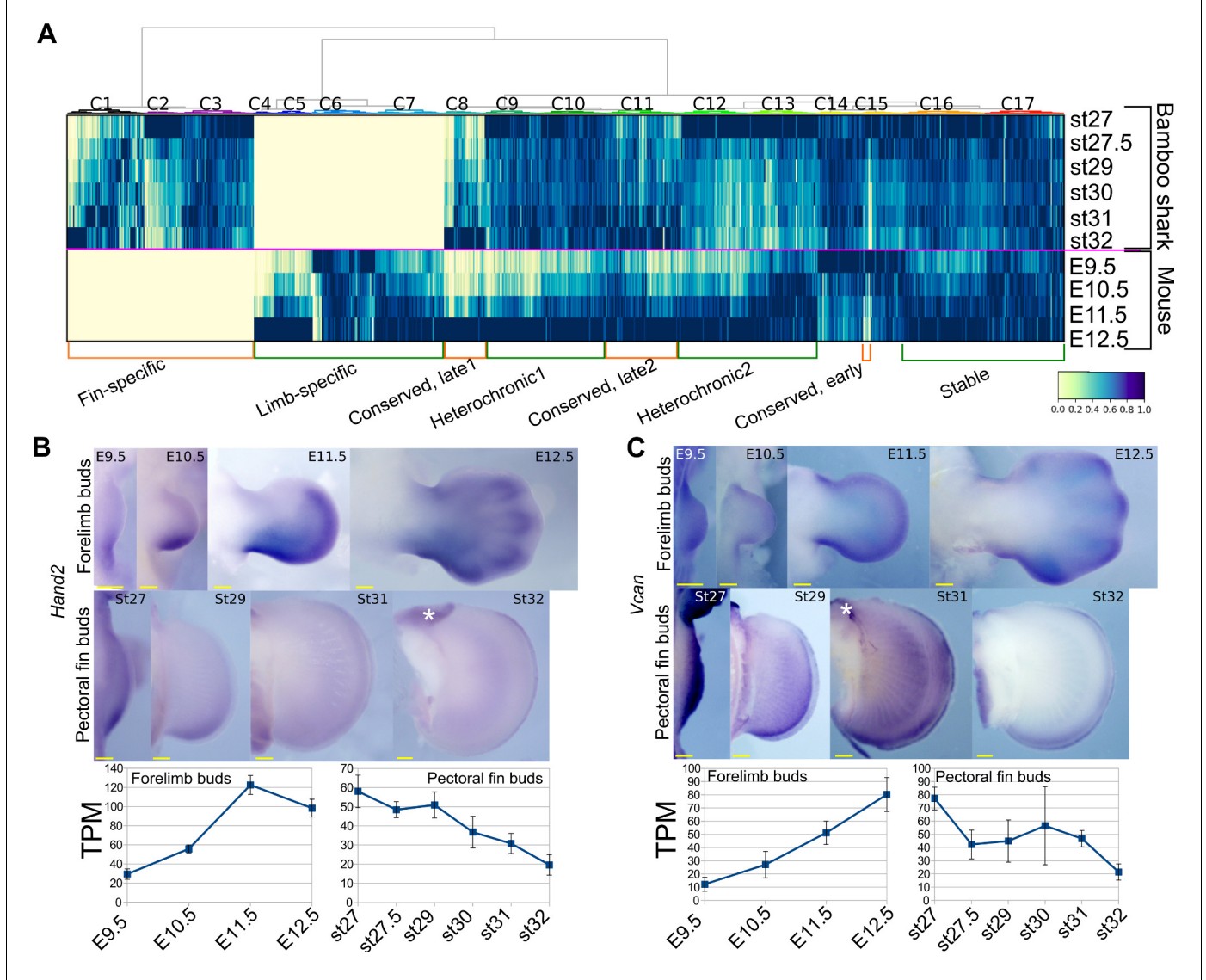

**Figure 2.** Detection of heterochronic gene expression between mouse limb buds and bamboo shark fin buds. (**A**) Clustering analysis of gene expression dynamics. Each column represents an ortholog pair between the bamboo shark and the mouse. Each row indicates scaled gene expression at a time point indicated to the right of the heat map. Values are scaled TPMs. (**B, C**) Whole-mount in situ hybridization of *Hand2* (**B**) and *Vcan* (**C**) as examples of the heterochronic genes detected in (**A**). Asterisks, background signals; scale bars, 200 µm. Error bars: SEM.

The online version of this article includes the following figure supplement(s) for figure 2:

**Figure supplement 1.** Other heterochronic genes.

*supplement 1C*). Although we cannot infer the ancestral state of the expression pattern, the overlapping functions of these genes may have allowed subfunctionalization of the signaling molecules of the AER during vertebrate divergence. In sum, we detected mass heterochronic shifts in gene expression between bamboo shark fin buds and mouse forelimb buds. In particular, a mechanism to maintain upregulation of the expression of genes involved in early fin development may have been either gained in the tetrapod lineages or lost in the cartilaginous fish lineages.

## Comparison of SHH signaling pathways in limb and fin buds

In tetrapod limbs, SHH controls growth and asymmetric gene expression along the anterior-posterior axis. Although previous studies have repeatedly implied a relatively delayed onset of *Shh* expression or a short signal duration in developing fins of several elasmobranch species (*Dahn et al., 2007*; *Sakamoto et al., 2009*; *Yonei-Tamura et al., 2008*), there has not been solid evidence to support such a delay due to the lack of systematic gene expression analysis and the poor staging system of these species. Because the heterochronic genes identified above include basic SHH target genes, such as *Ptch1* and *Gli1*, we reexamined the expression dynamics of *Shh* and its target genes in mouse limb and bamboo shark fin buds. Because HOX genes are the upstream factors relative to *Shh* transcription (*Zeller et al., 2009*), we used them as a potential reference for developmental time. We first found that *Shh* transcription was present by the earliest stages examined in both bamboo shark fin and mouse limb buds, and it peaked when the transcription level of *Hoxd9* and *Hoxd10* was highest, suggesting that there was no apparent heterochrony in *Shh* transcription timing at least between these two species (red rectangles in *Figure 3A and B*). In contrast, SHH target genes, such as *Ptch1/2*, *Gli1*, *Gremlin*, and *Hand2* (*Vokes et al., 2008*), did show a

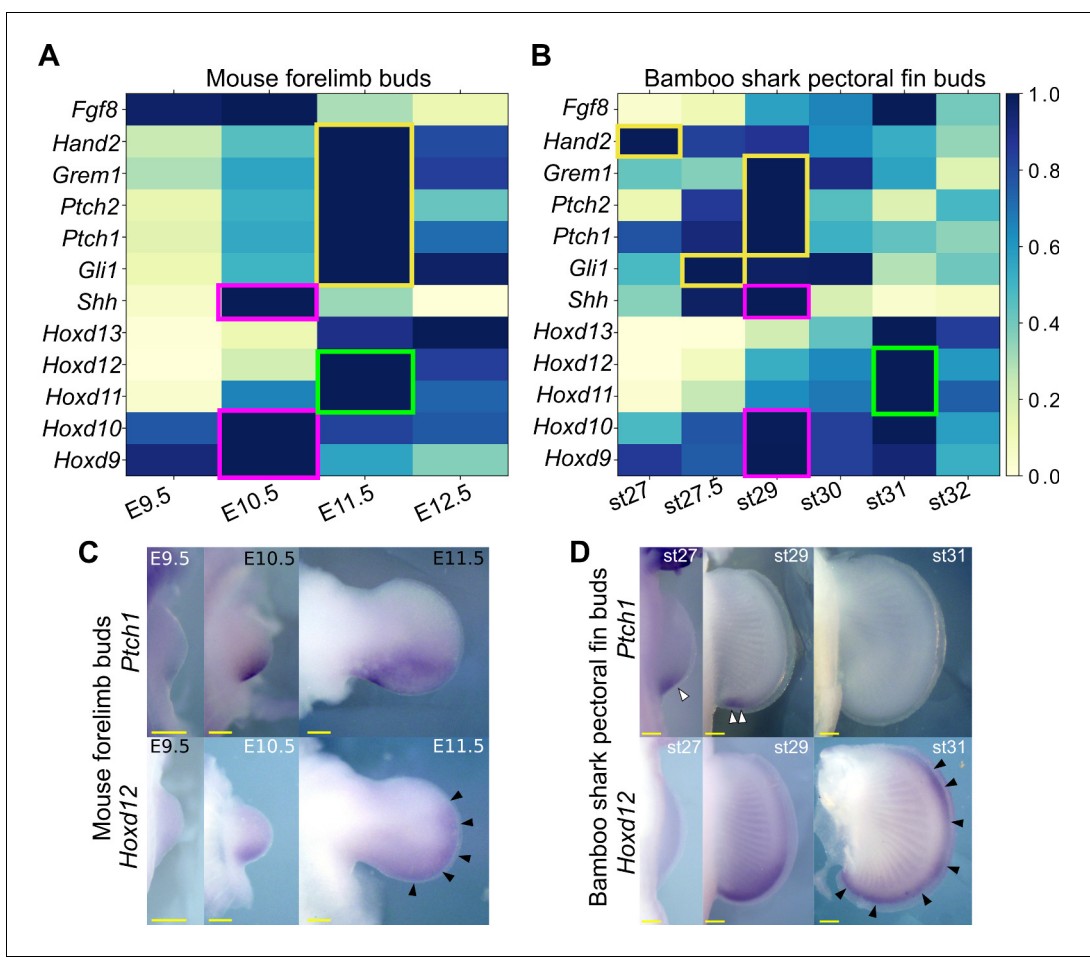

**Figure 3.** *Shh* pathway in mouse limb buds and bamboo shark fin buds. (A, B) Scaled expression of *Shh* and related genes in mouse limb buds (A) and bamboo shark fin buds (B), respectively. The rectangles indicate the expression peaks of *Shh*, *Hoxd9*, and *Hoxd10* (magenta), Shh target genes (yellow) and *Hoxd11* and *Hoxd12* (green). (C, D) Whole-mount in situ hybridization of *Ptch1* and *Hoxd12* in mouse limb buds (C) and bamboo shark fin buds (D); scale bars, 200 µm. White arrowheads in (D) indicate restricted expression of *Ptch1* in bamboo shark fin buds. Black arrowheads in (C and D) indicate anteriorly extended expression of *Hoxd12*.

The online version of this article includes the following figure supplement(s) for figure 3:

**Figure supplement 1.** The temporal dynamics of the Shh pathway based on intact TPM values.

relatively extended period of expression in mouse limb buds as compared with their expression in bamboo shark fin buds. Namely, whereas the expression peak of SHH target genes was concurrent with that of *Shh* in the bamboo shark fin bud, these SHH target genes were highly expressed in E11.5 limb buds, which is one day later than the *Shh* expression peak (yellow rectangles in *Figure 3A and B*; see *Figure 3—figure supplement 1* for intact TPM values). This timing difference is also apparent when comparing the expression peak of *Hoxd11* and *Hoxd12*, which was concurrent with that of SHH target genes in mouse limb buds, but came after downregulation of SHH target genes in bamboo shark fin buds (green rectangles in *Figure 3A and B*). To confirm this observation, we performed whole-mount in situ hybridization for *Ptch1* and *Hoxd12* in mouse limb buds and bamboo shark fin buds. As previously reported (*Lewis et al., 2001*; *Zákány et al., 2004*), mouse limb buds showed a clear expansion of the expression domain of *Ptch1* (upper panel in *Figure 3C*) from E10.5 to E11.5, which is accompanied by the anterior extension of the *Hoxd12* expression domain (black arrowheads in *Figure 3C*). In contrast, *Ptch1* was expressed in the posterior domain of bamboo shark fin buds at st. 29 (white arrowheads in *Figure 3D*), but was substantially downregulated by st. 31, whereas the *Hoxd12* expression domain extended anteriorly at this stage (black arrowheads in *Figure 3D*). These results were roughly consistent with the RNA-seq data. We cannot completely reject the possibility that this timing difference is due to the different physical time-resolution of data sampling between these species (six time points over 20 days in the bamboo shark and four time points over 4 days in the mouse). However, given that this data set captured the similar expression dynamics of HoxA/D clusters between these species (*Figure 1D*; also see *Figure 4C*) as well as the differentiation dynamics of myocytes and chondrocytes (*Figure 1—figure supplement 8*), these results quite likely represent an interesting difference in the transcriptional regulation of SHH downstream genes between fins and limbs.

## Hourglass-shaped conservation

Several studies have reported a temporally heterogeneous diversification of embryonic transcriptomes, such that the middle stages are more conserved than early or late stages (e.g. *Irie and Kuratani, 2011*; *Kalinka et al., 2010*; *Levin et al., 2012*). These observations are considered to support the notion of the developmental hourglass (or egg timer), which has been proposed to explain the morphological similarity of mid-stage embryos based on developmental constraints, such as strong interactions between tissues or Hox-dependent organization of the body axis (*Duboule, 1994*; *Raff, 1996*). In addition, a previous transcriptomic analysis reported that the late stage of mammalian limb development has experienced relatively rapid evolution (*Maier et al., 2017*). To examine which developmental stages of fins and limbs are conserved, we calculated the distance between the fin and limb transcriptome data. As a result, four different distance methods that we examined consistently indicated that the limb bud at E10.5 and the fin buds at st27.5–30 tended to have a relatively similar expression profile (*Figure 4A* for a Euclidean distance measure and *Figure 4—figure supplement 1* for other types of distance measures). In addition, the transcriptomic profile of all the stages of examined fin buds showed the highest similarity to that of E10.5 limb bud (*Figure 4B*). Therefore, the mid-stages of limb and fin buds tend to be conserved over 400 million years of evolution.

To find factors that underlie the mid-stage conservation, we analyzed *Hox* genes, which were proposed to be responsible for the developmental hourglass (*Duboule, 1994*). We found that the comparison of only *Hox* gene expression did not reproduce the hourglass-shaped conservation (*Figure 4C*), suggesting that other mechanisms constrain the middle stage of development. We further performed principal component analysis (PCA) of gene expression profiles to identify genes responsible for the hourglass-shaped conservation. The first component, PC1, distinguished transcriptome data mostly by species differences (*Figure 4D*). In contrast, PC2 was correlated with the temporal order of mouse limb buds (*Figure 4D*). PC2 was also weakly correlated with the temporal order of bamboo shark fin buds except at st27 (*Figure 4D*), but PC3 showed a clearer correlation (*Figure 4E*). These three components were mostly sufficient to reproduce the mid-stage conservation in *Figure 4A* (*Figure 4—figure supplement 2A* for the ratio of explained variables and 2B for the Euclidean distance measure). Interestingly, the plot with PC2 and PC3 roughly mirrored the hourglass-shaped conservation because the earliest and latest stages were placed more distantly than the middle stages in this representation (*Figure 4E*). Indeed, the major loadings of PC2 consisted of the conserved late expressed genes (C8) and the heterochronically regulated genes (C9 and C12)

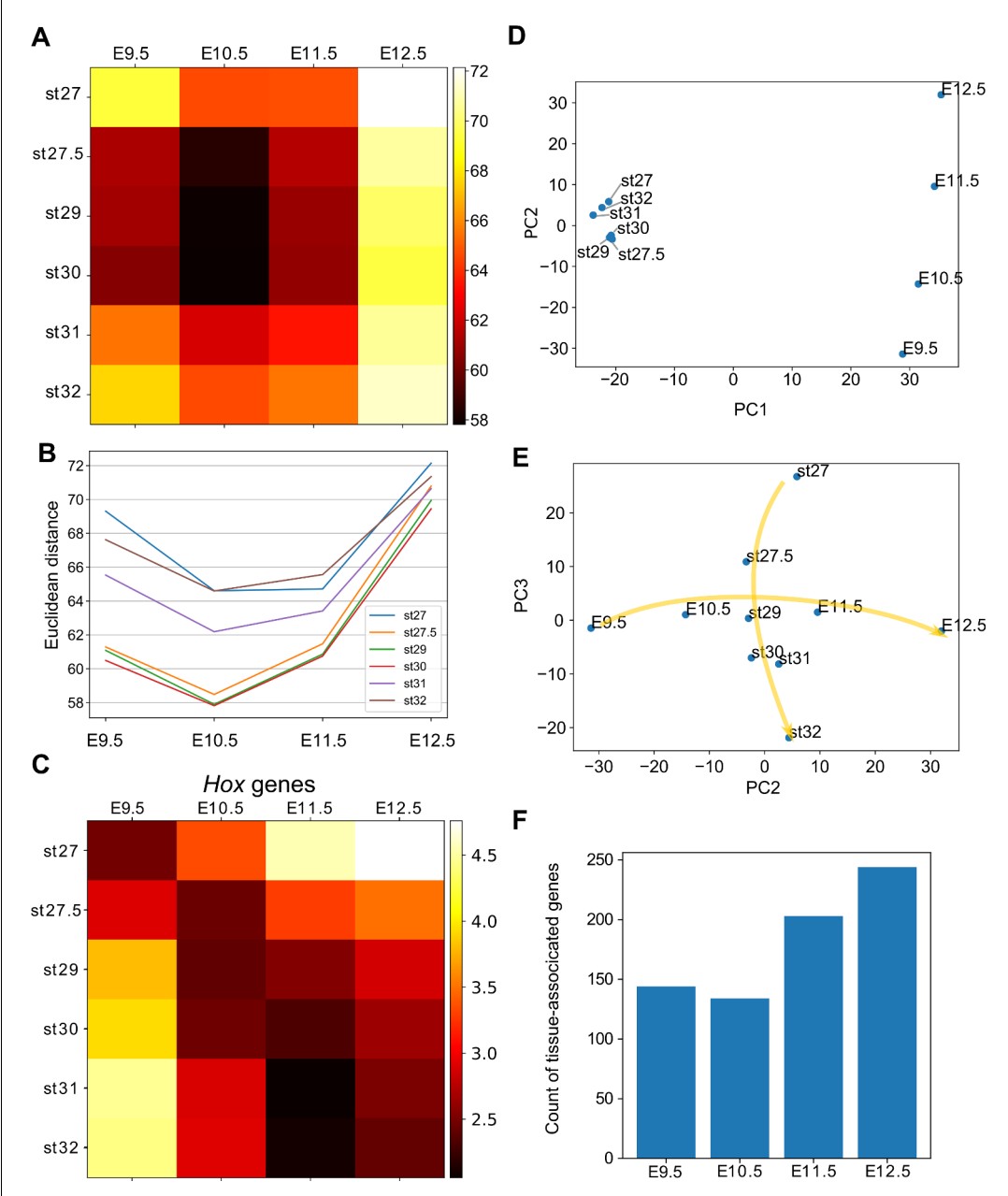

**Figure 4.** Hourglass-shaped conservation of the transcriptome profile between fins and limbs. (A) Euclidean distances of the transcriptome profiles. Every combination of time points of bamboo shark fin buds and mouse limb buds is shown. The darker colors indicate a greater similarity between gene expression profiles. (B) A line plot of the Euclidean distances shown in (A). The x axis indicates the mouse limb stages, and the y axis is the Euclidean distance. (C) The same as (A) except that only *Hoxd* genes are included. (D, E) Scatter plots of the first and second principal components (D) and of the second and third components (e). Arrows in (E) indicate the time-order of transcriptome data. (F) Count of tissue-associated genes expressed in mouse forelimb buds. Genes with $0.65 \leq$ entropy were counted.

The online version of this article includes the following figure supplement(s) for figure 4:

**Figure supplement 1.** Confirmation analyses of the transcriptome comparison.

**Figure supplement 2.** Additional PCA data and counts for stage- and tissue-associated genes.

identified in *Figure 2A* (see *Table 2* for the top 25 genes of PC2). Similarly, PC3 consisted of the conserved early genes (a part of C15) and the heterochronically regulated genes (C12 and C13; see *Supplementary file 9* for the loadings of PC3 and others), suggesting that the presence of heterochronically regulated genes may at least partly contribute to the mid-stage conservation and the distant relationship between the early/late stages of fins and limbs. These results indicate that the mass heterochronic shift in gene expression, at least in part, contributes to the long distances between early- and late-stage expression profiles (*Figure 4E*).

Because a recent report suggests that pleiotropy of genes is related to hourglass-shaped conservation (*EXPANDE Consortium et al., 2017*), we counted the number of genes with stage- or tissue-specific expression. Consistent with the previous report (*EXPANDE Consortium et al., 2017*), we detected a relatively low number of stage-associated genes during the middle stages of mouse forelimb and bamboo shark fin development (*Figure 4—figure supplement 2C*). To evaluate the tissue specificity of genes, we first calculated Shannon entropy of gene expression patterns by analyzing RNA-seq data from 71 mouse tissues as released by the ENCODE project (*Davis et al., 2018*; *Supplementary file 10* for the list of RNA-seq data). Namely, genes expressed only in a few tissues score lower with respect to entropy (thus, these genes are more specific). We counted genes with $1.0 \geq$ TPM and $0.65 \leq$ entropy and, again, found that the number of tissue-associated genes was relatively low at E10.5 (*Figure 3F*). Together, these results indicate an inverse correlation between the hourglass-shaped conservation and the number of tissue- and stage-specific genes.

**Table 2.** PCA loadings.

**Loading axis: PC2**

| Gene symbol | Cluster name | Loading |
| --- | --- | --- |
| TRHDE | C8 | 0.31 |
| PAX9 | C11 | 0.3 |
| COL9A2 | C8 | 0.3 |
| RTN4R | C8 | 0.3 |
| APC2 | C9 | 0.3 |
| CNMD | C8 | 0.29 |
| HOXD13 | C8 | 0.29 |
| FAM69C | C8 | 0.29 |
| WFIKKN2 | C8 | 0.29 |
| HOXA13 | C8 | 0.29 |
| LRRN3 | C12 | 0.29 |
| HPSE2 | C9 | 0.29 |
| SERPINB1A | C11 | 0.29 |
| CDKN2B | C8 | 0.28 |
| LTBP1 | C8 | 0.28 |
| CDH19 | C8 | 0.28 |
| PDZD2 | C8 | 0.28 |
| NLGN3 | C9 | 0.28 |
| MATN1 | C8 | 0.28 |
| MYOD1 | C8 | 0.28 |
| TSPAN11 | C12 | 0.28 |
| SERINC2 | C9 | 0.28 |
| FYB | C8 | 0.28 |
| KIF1A | C8 | 0.28 |
| COL9A3 | C8 | 0.28 |

## Open chromatin region (OCR) conservation

Next, we systematically identified putative gene regulatory sequences involved in mouse limb development and sought a possible cause for the hourglass-shaped conservation in gene regulatory sequences. To this end, we applied ATAC-seq, which detects OCRs (putative active regulatory sequences), to time-series of forelimb buds at E9.5–E12.5 with three replicates. First, as a positive control, we found that ATAC-seq peaks that were determined by MACS2 peak caller covered 10 of 11 known limb enhancers of the HoxA cluster (*Figure 5A* and *Figure 5—figure supplement 1*), suggesting a high coverage of true regulatory sequences. Consistently, our ATAC-seq data showed relatively high scores for a quality control index, fraction of reads in peaks (FRiP), as compared with data downloaded from the ENCODE project (*Davis et al., 2018*; *Figure 5B*). Next, to examine evolutionary conservation, we performed BLASTN (*Camacho et al., 2009*) for the sequences in the ATAC-seq peaks against several vertebrate genomes. Reinforcing the result of the transcriptome analysis, we

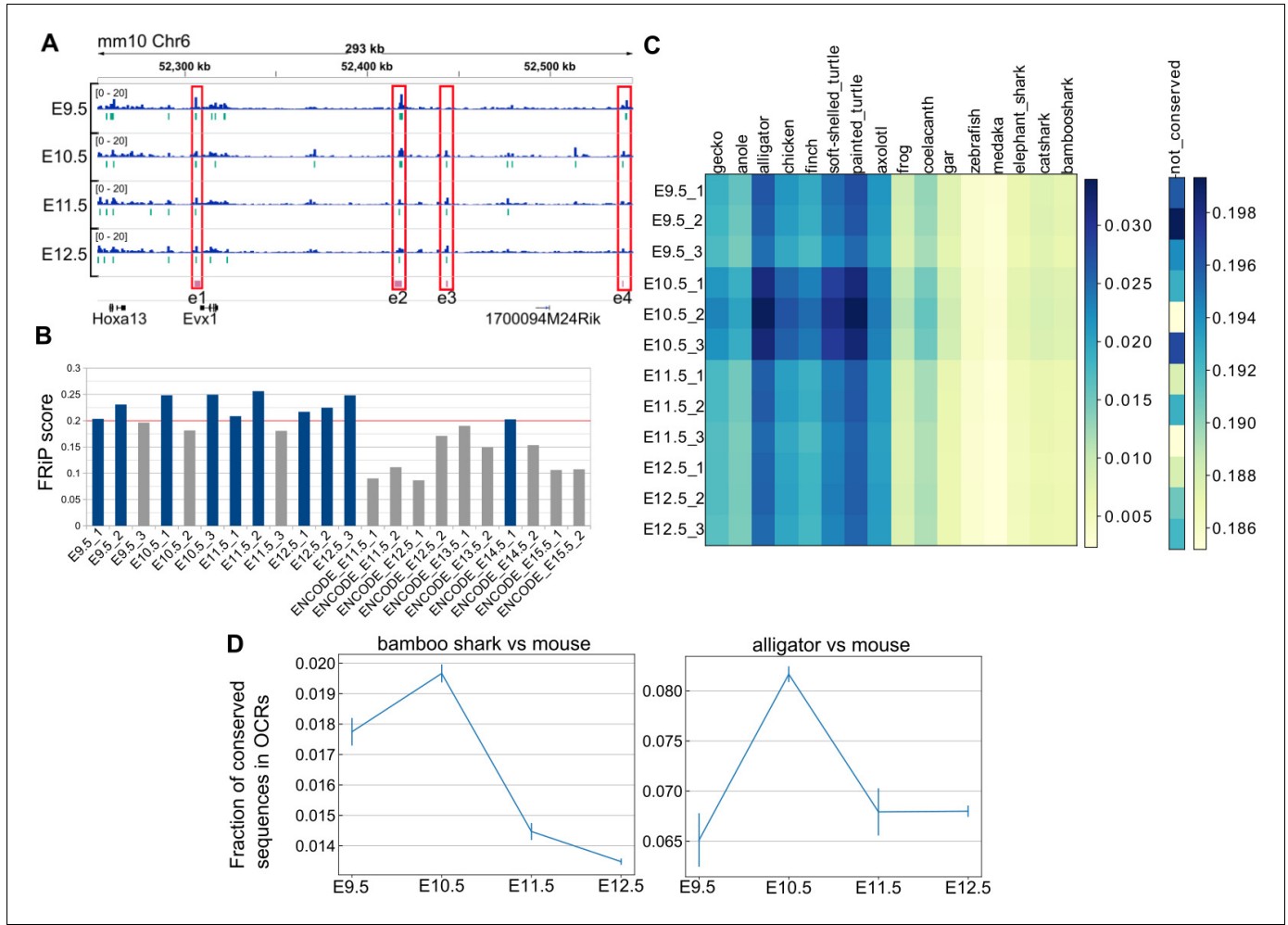

**Figure 5.** Hourglass-shaped conservation of OCRs in mouse limb development. (**A**) ATAC-seq signals in the enhancer regions of the HoxA cluster. e1 to e4, known limb enhancers. Green vertical lines below the signals, peak regions determined by MACS2. (**B**) Comparison of a quality index, FRiP, for ATAC-seq data. Blue bars are samples with a FRiP score >0.2. The number in the end of the label name indicates the replicate number. (**C**) Conservation analysis of sequences in ATAC-seq peaks with BLASTN. The values to the right of each graph indicate the fraction of conserved sequences in the total peak regions. The common name of each genome sequence is indicated above the graph. The not-conserved heatmap indicates the fraction of sequences that were not aligned to any genome sequences and thus serve as a negative control. (**D**) Temporal changes of sequence conservation frequency in ATAC-seq peaks with LAST. Error bars: SEM.

The online version of this article includes the following figure supplement(s) for figure 5:

**Figure supplement 1.** ATAC-seq quality control.

**Figure supplement 2.** Conservation measures of OCRs.

found that evolutionarily conserved sequences were most accessible at E10.5 (*Figure 5C*). To confirm this result, we also used a different alignment algorithm, LAST (*Kiełbasa et al., 2011*) with the bamboo shark and the alligator (*Green et al., 2014*) genomes. Alignment results for both analyses consistently indicated that the OCRs of E10.5 forelimb bud more frequently contained conserved sequences relative to those of other time points (*Figure 5D*; see *Figure 5—figure supplement 2A and B* for the absolute counts of conserved sequences). Therefore, activation of conserved gene regulatory sequences may be one of the proximate causes for the hourglass-shaped conservation of fin and limb transcriptome data.

## Temporal dynamics of open chromatin domains

To further characterize the ATAC-seq peaks, we next performed a clustering analysis. Using one of the three replicates for each stage, we collected the summits of peaks and the surrounding 1400 bp and carried out hierarchical clustering, which resulted in eight clusters (C1–C8; *Figure 6A*) that consisted of broad (C1 and C2), E11.5/E12.5-specific (C3 and C4), stable (C5 and C6), E10.5-specific (C7), and E9.5-specific (C8) peaks. The overall clustering pattern was reproducible by other

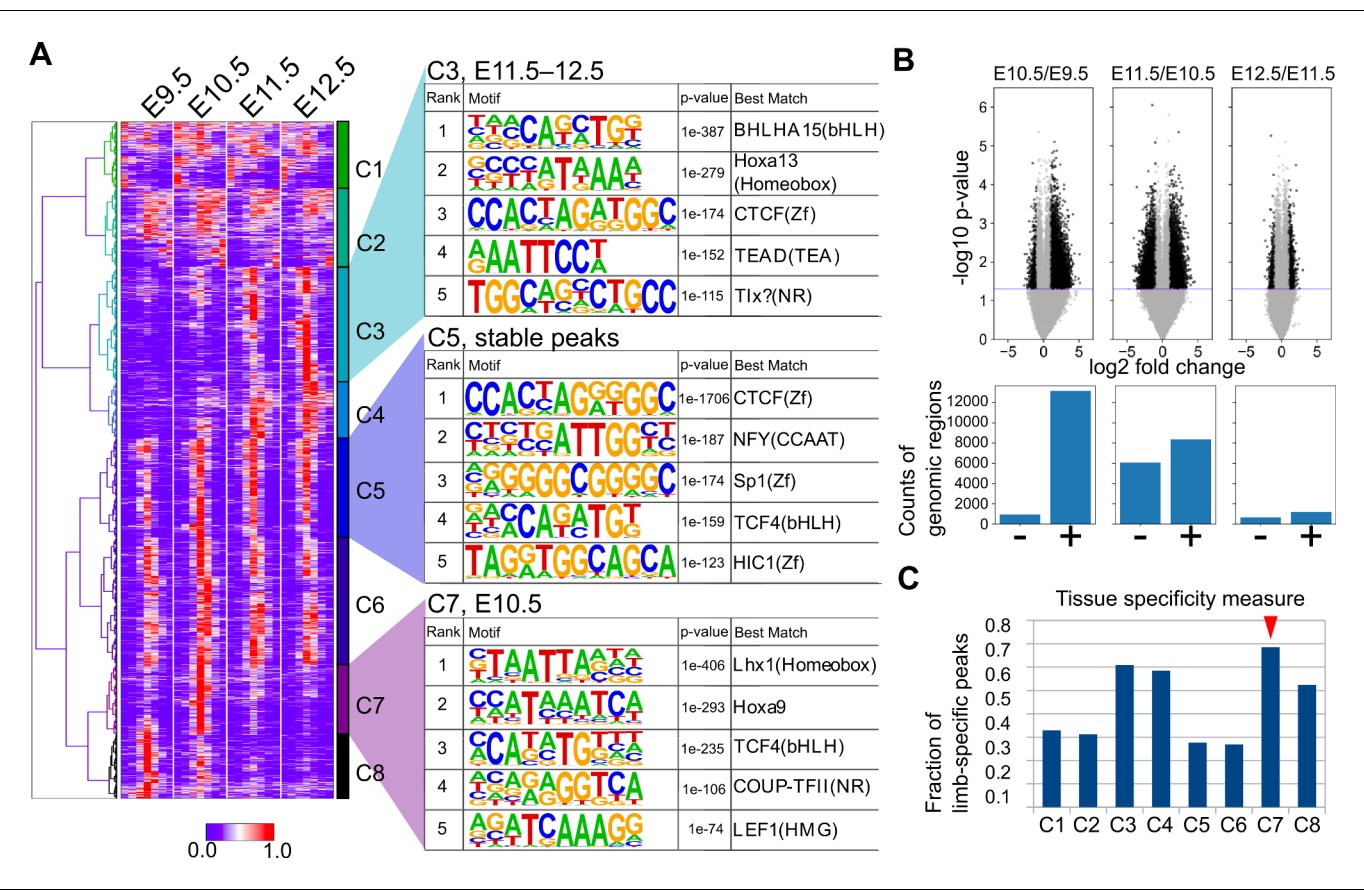

**Figure 6.** Temporal dynamics of OCRs during mouse limb development. (**A**) The heatmap (left) shows whole-genome clustering of ATAC-seq peaks. Each row indicates a particular genome region with a length of 1400 bp. Columns indicate developmental stages. C1−C8 are cluster numbers. The motifs (right) show the rank of enriched motifs in the sequences of each cluster. (**B**) Top, volcano plots of ATAC-seq signals between indicated stages (p-values, two-sided Student's t-test). Bottom, the counts of differential signals (black dots in the top panel). + and − are genomic regions with increased or decreased signals, respectively. (**C**) The fraction of limb-specific OCRs for each cluster.

The online version of this article includes the following figure supplement(s) for figure 6:

**Figure supplement 1.** Clustering analyses of ATAC-seq peaks with different replicates.
**Figure supplement 2.** Analysis of enrichment for known motifs in ATAC-seq peaks.
**Figure supplement 3.** De novo motif discoveries and known motif enrichment analysis of ATAC-seq peaks with an alternative background.
**Figure supplement 4.** De novo motif discoveries of ATAC-seq peaks.
**Figure supplement 5.** Counts of accessible motifs at each stage.

combinations of replicates if its FRiP was ≥0.20 (*Figure 6—figure supplement 1*). Consistent with the above conservation analysis, E10.5-specific peaks frequently overlapped conserved sequences (*Figure 5—figure supplement 2C and D*).

To characterize the regulatory features of the clusters, we performed motif analysis in each cluster using HOMER (*Heinz et al., 2010*). First, it was convincing that stable peaks (C5 and C6) were enriched for the CTCF binding motif both in de novo motif discovery (*Figure 6A*) and known motif enrichment analysis (*Figure 6—figure supplement 2*), which is a major regulator of three-dimensional genomic structure. This result was consistent whether random genomic regions or other peak regions were used for the background (*Figure 6—figure supplement 3*). In addition, E11.5/E12.5-specific peak C3 was enriched for the HOX13 motif (*Figure 6A*), which was consistent with the increase in the expression of 5′ *Hox* genes (*Figure 1D*). C4 was also enriched for motifs similar to those of C3, but the HOX13 motif was detected only in known motif enrichment analysis (compare *Figure 6—figure supplements 2* and *3*). The enrichment of the HOX9 motif in E10.5-specific peaks (C7) was also consistent with our RNA-seq data, in which *Hoxd9* and *Hoxa9* expression levels peaked at E10.5 (*Figure 1D*). Interestingly, in E10.5-specific peaks (C7), the LHX1-binding motif was ranked at the top of the motif enrichment list the closely related transcription factors *Lhx2*, *Lhx9*, and *Lmx1b* are required to mediate a signaling feedback loop between ectoderm and mesenchyme in limb development (*Tzchori et al., 2009*). C8 was enriched for motifs similar to those in C7 (e.g. COUP-TFII), but the top-ranked transcription factor in the de novo motif discovery analysis was VSX2, which has a very similar binding sequence to the LHX motif (*Figure 6—figure supplement 4*). The LHX motif was top-ranked in C8 for the known motif enrichment analysis (*Figure 6—figure supplement 2*). For a better understanding of the dynamics of transcription factor motifs, we counted the average number of the above detected motifs within the OCRs of each stage, which revealed a transitional increase in LHX and HOX9 motifs at E10.5 and a gradual increase in the motifs detected in C3 over the developmental stages (*Figure 6—figure supplement 3*). In addition, Gene Ontology (GO) analysis for the peaks in each cluster revealed that the constitutively accessible peaks (C5, C6) were closely located to genes annotated with 'cellular components' (*Supplementary file 11*). Interestingly, the dynamically regulated peaks (C3, C4, C7, C8) were associated with genes with 'developmental process', 'multicellular organism development', and 'anatomical structure morphogenesis' (*Supplementary file 11*), suggesting that these dynamic OCRs regulate developmental genes. Together, these results suggest that there are E10.5-specific transient OCRs that exhibit several characteristics including their evolutionary conservation, the presence of LHX and HOX9 motifs and a close relation with developmental genes.

To confirm the results from the above clustering analysis, we also determined the genomic regions that showed a statistically significant increase or decrease in the ATAC-seq signal within a day by using all replicates. As a result, ATAC-seq signals were most increased during the transition from E9.5 to E10.5 in the mouse limb bud. From E10.5 to E11.5, the total number of decreased and increased signals was highest, indicating that the OCR landscape was most dynamically changing at E10.5 (*Figure 6B*). In contrast, relatively few significant changes were observed from E11.5 to E12.5. Thus, in contrast to the transcriptome analysis, stage-specific gene regulatory sequences are likely to be most accessible at E10.5. Moreover, by comparing the peaks of each cluster identified above with ATAC-seq peaks of other cells and tissues released by the ENCODE project (*Davis et al., 2018*; *Supplementary file 10* for the full list of cells and tissues), we discovered that the C7 cluster (E10.5-specific peaks) contained more peaks that did not overlap with those of other cells and tissues. Again, in contrast to the transcriptome analysis, the data suggest that gene regulatory sequences that are accessible only at E10.5 tend to be limb-specific (*Figure 6C*). Taken together, these analyses revealed a unique regulatory landscape of forelimb buds at E10.5, which is enriched for evolutionarily conserved stage-specific and tissue-specific OCRs.

## Discussion

In this work, we applied transcriptomics and chromatin accessibility analysis to systematically study genetic changes that differentiate fins from limbs. Because of the slow sequence evolution and the embryo availability of the bamboo shark, we were able to compare transcriptional regulation of genes with high accuracy and found both heterochronic shifts and hourglass-shaped conservation of

transcriptional regulation between fin and limb development. Here, we discuss the interpretations, limitations, and implications of these results.

Our time-series transcriptome data indicated that a remarkable number of genes that exhibit the highest expression during the late stages of mouse limb bud development are decreased during the late stages of bamboo shark fin development (*Figure 2*). The simplest hypothesis for this mass heterochronic shift is that the later stages of limb development gained expression of one or a few upstream transcription factor(s) or signaling molecules that collectively regulate this group of genes. Interestingly, we also observed relatively extensive expression of the downstream targets of the SHH signaling pathway in mouse limb buds, as compared with bamboo shark fin buds (*Figure 3*). Because SHH-independent regulation of its target genes through the GLI3-HOX complex was previously reported (*Chen et al., 2004*), the mismatch between the peak expression of *Shh* and its target genes may be caused by such SHH-independent regulatory mechanisms that are absent in bamboo shark fin development. Given that direct and genetic interactions of GLI3 and HOX have a significant impact on autopod formation, the emergence of this interaction may be a key component of the mass heterochronic shift and the acquisition of autopod-related developmental regulation in the tetrapod lineages. However, because we compared only two species, it is equally possible that the late stages of shark fin development lost this SHH-independent gene regulation. Alternatively, given that the evolutionary distance between these two species is >400 million years, it is also possible that every one of these genes independently shifted their expression to the later stages of limb development or to the early stages of shark fin development. Further taxon sampling and functional analyses will reveal the relation between the mass heterochronic shift and the emergence of the autopod.

Related to the potential changes in regulation of SHH target genes, by analyzing catshark fin buds, we previously proposed that the expression domains of genes that are positively regulated by SHH might have expanded anteriorly during the fin-to-limb transition (*Onimaru et al., 2015*). We speculated that this expression changes may be linked to the loss of pro- and mesopterygial elements. Recently, this hypothesis was partially supported by another group who compared the gene expression pattern of lungfish and cichlid fin development (*Woltering et al., 2020*), where lungfish fin buds seem to exhibit an intermediate condition between non-sarcopterygian fish fins and tetrapod limbs in terms of gene expression distribution along the anterior-posterior axis. This group particularly emphasized that the absence and presence of the dynamics of the anterior expansion of *Hoxd13* expression correlate with the difference between the metapterygial morphologies of lungfish and tetrapods (also see *Johanson et al., 2007* for a conflicting report). However, the significance of changes in *Hoxd13* expression remains unclear because of the following two reasons: (a) *Hoxd13* expression pattern seems to quite vary among species—the anterior expansion of *Hoxd13* expression has been observed in the fin buds of the little skate, the small-spotted catshark, and *Polyodon* (*Davis et al., 2007*; *Freitas et al., 2007*; *Nakamura et al., 2015*), while not in those of zebrafish and cichlids (*Ahn and Ho, 2008*; *Woltering et al., 2020*) and (b) in fish fins, the expression domain of *Hoxa13,* whose function is mostly redundant with *Hoxd13,* commonly spans from anterior to posterior regions in fish fin buds like as tetrapod limbs (*Davis et al., 2007*; *Freitas et al., 2007*; *Nakamura et al., 2016*). Therefore, while changes in *Hoxd13* expression domain are likely to contribute to some degree of anatomical diversity, their impact is questionable in the context of the fin-to-limb transition. Nevertheless, our previous study and Woltering et al. commonly suggest that the anterior expansion of gene expression domains is likely associated with the substantial anatomical changes during the fin-to-limb transition. As discussed above, we speculate that the mass heterochronic shifts that we observed in the present study may be related to the gain of SHH-independent regulation of its target genes. Therefore, whether the anterior expansion of SHH-target gene expression is related to the mass heterochronic shifts will be one of the interesting questions to address in the future.

We observed that gene expression profiles are most highly conserved between bamboo shark fin buds at st. 27.5–30 and mouse forelimb buds at E10.5 (*Figure 4*). Consistent with this result, our chromatin accessibility analysis reveals that OCRs at E10.5 tend to contain evolutionarily conserved sequences (*Figure 5*). Whereas transcriptomic conservation during the middle of embryonic development has been reported by many groups using different species (e.g. *Irie and Kuratani, 2011*; *Kalinka et al., 2010*), analysis of regulatory sequence conservation during embryonic development has been either incomplete or controversial. For example, by analyzing histone acetylation marks on several developing organs in mouse embryos, *Nord et al., 2013* proposed regulatory sequences

active at E11.5 are exposed by the highest evolutionary constraint. However, they used stem cell lines as the substitutes for organs at early stages. Another study showed that genes expressed at the segmentation stage of zebrafish embryos tended to be surrounded by highly conserved noncoding sequences (*Piasecka et al., 2013*). Although their results are in line with our present study as discussed below, they did not show that these highly conserved non-coding sequences were indeed active at the segmentation stage. In addition to these studies, there is a conflicting observation that early, instead of middle, embryonic stages tend to be regulated by conserved OCRs (*Uesaka et al., 2019*). Therefore, our present study is the first to convincingly show a clear correlation of conservation status between transcriptomic data and OCRs. Our results suggest that evolutionary constraints on the gene regulatory apparatus are present during the middle stage of fin and limb development. What drives the hourglass-shaped conservation is still under debate. Interestingly, we found that stage- and tissue-specific OCRs were enriched in this conserved period, during which a relatively low number of stage- and tissue-specific genes were expressed (*Figure 6*). These quite contrasting observations imply that the mid-stage limb development is enriched for pleiotropic genes controlled by multiple tissue-specific enhancers, including limb-specific ones, rather than by constitutive promoters that often regulate housekeeping genes. Therefore, we speculate that, at least in the case of limb development, complex regulatory sequences that execute spatiotemporally specific transcriptional controls over pleiotropic genes constrain the evolvability of this particular period of morphogenesis, probably due to the vulnerability of complex regulation to genetic mutations.

In conclusion, the present study provides insights for the evolutionary origin of gene regulation that differentiates fins from limbs. In particular, comparative transcriptional analyses prompted us to hypothesize that mass heterochronic shifts of gene expression may have occurred during the fin-to-limb evolution. In addition, both transcriptome and open chromatin data point to an evolutionary constraint during mid-stage limb development, likely owing to gene regulatory complexity. Although these hypotheses require further taxon sampling and experimental tests, this study opens up new prospects for understanding not only the genetic basis of the fin-to-limb transition but also the general nature of morphological evolution.

# Materials and methods

**Key resources table**

| Reagent type (species) or resource | Designation | Source or reference | Identifiers | Additional information |
|---|---|---|---|---|
| Gene (*Mus musculus*) | Hand2 | ENSEMBL | ENSMUST00000040104.4 | N/A |
| Gene (*Mus musculus*) | Vcan | ENSEMBL | ENSMUST00000109546.8 | N/A |
| Gene (*Mus musculus*) | Aldh1a2 | ENSEMBL | ENSMUST00000034723.5 | N/A |
| Gene (*Mus musculus*) | Ptch1 | ENSEMBL | ENSMUST00000192155.5 | N/A |
| Gene (*Mus musculus*) | Hoxd12 | ENSEMBL | ENSMUST00000001878.5 | N/A |
| Gene (*Chiloscyllium punctatum*) | Hand2 | ENSEMBL | Chipun0004250/g4250.t1/ TRINITY_DN85524_c0_g1_i1 | N/A |
| Gene (*Chiloscyllium punctatum*) | Vcan | This paper | Chipun0003941/g3941.t1/ TRINITY_DN95522_c0_g1_i8 | N/A |
| Gene (*Chiloscyllium punctatum*) | Hoxd12 | This paper | Chipun0005654/g5654.t1/ TRINITY_DN85970_c0_TRINITYg1_i1 | N/A |
| Gene (*Chiloscyllium punctatum*) | Ptch1 | This paper | Chipun0003320/g3320.t1/ TRINITY_DN92499_c0_g1_i3 | N/A |
| Gene (*Chiloscyllium punctatum*) | Aldh1a2 | This paper | Chipun0010503/g10503.t1/ TRINITY_DN81423_c0_g1_i1 | N/A |
| Strain, strain background (*Mus musculus*) | C52BL/6 | Laboratory for Animal Resources and Genetic Engineering RIKEN, | N/A | N/A |

*Continued on next page*

*Continued*

| Reagent type (species) or resource | Designation | Source or reference | Identifiers | Additional information |
|---|---|---|---|---|
| Antibody | Anti-Digoxigenin-AP, Fab fragments (Sheep) | Millipore Sigma | Cat# 11093274910 | polyclonal (1:4000) |
| Sequence-based reagent | *Mus musculus* Hand2 forward primer | This paper | PCR primers | ACCAAACTC TCCAAGA TCAAGACACTG |
| Sequence-based reagent | *Mus musculus* Hand2 reverse primer | This paper | PCR primers | TTGAATACTTACAA TGTTTACACCTTC |
| Sequence-based reagent | *Mus musculus* Vcan forward primer | This paper | PCR primers | TGCAAAGATGG TTTCA TTCAGCGACAC |
| Sequence-based reagent | *Mus musculus* Vcan reverse primer | This paper | PCR primers | ACACG TGCAGAGACC TGCAAGATGCTG |
| Sequence-based reagent | *Mus musculus* Aldh1a2 forward primer | This paper | PCR primers | ACCGTGTTC TCCAACGTCACTGA TGAC |
| Sequence-based reagent | *Mus musculus* Aldh1a2 reverse primer | This paper | PCR primers | TCTGTCAGTAACAG TATGGAGAGCTTG |
| Sequence-based reagent | *Mus musculus* Hoxd12 forward primer | This paper | PCR primers | CTCAACTTGAACA TGGCAGTGCAAGTG |
| Sequence-based reagent | *Mus musculus* Hoxd12 reverse primer | This paper | PCR primers | AGCTCTAGCTAGGC TCCTGTTTCATGC |
| Sequence-based reagent | *Mus musculus* Ptch1 forward primer | This paper | PCR primers | GGGAAGGCAGTTCA TTGTTACTGTAACTG |
| Sequence-based reagent | *Mus musculus* Ptch1 reverse primer | This paper | PCR primers | TGTAATACGACTCAC TATAGGTCAGAAGC TGCCACACACAGG-CATGAAGC |
| Sequence-based reagent | *Chiloscyllium punctatum* Hand2 forward primer | This paper | PCR primers | ACCAGCTACATTGCC TACCTCATGGAC |
| Sequence-based reagent | *Chiloscyllium punctatum* Hand2 reverse primer | This paper | PCR primers | CACTTG TTGAACGGAAG TGCACAAGTC |
| Sequence-based reagent | *Chiloscyllium punctatum* Vcan forward primer | This paper | PCR primers | AGCTTGGGAAGA TGCAGAGAAGGAA TG |
| Sequence-based reagent | *Chiloscyllium punctatum* Vcan reverse primer | This paper | PCR primers | AGAGCAGCTTCACAA TGCAGTCTCTGG |
| Sequence-based reagent | *Chiloscyllium punctatum* Aldh1a2 forward primer | This paper | PCR primers | TTGAACTTGTAC TAAGTGGTATCGCTG |
| Sequence-based reagent | *Chiloscyllium punctatum* Aldh1a2 reverse primer | This paper | PCR primers | AGGATGTGAACA TTAGGCTGACCTCAC |

*Continued on next page*

*Continued*

| Reagent type (species) or resource | Designation | Source or reference | Identifiers | Additional information |
|---|---|---|---|---|
| Sequence-based reagent | *Chiloscyllium punctatum Hoxd12* forward primer | This paper | PCR primers | GCCAGTA TGCAACAGATCCTC TGATGG |
| Sequence-based reagent | *Chiloscyllium punctatum Hoxd12* reverse primer | This paper | PCR primers | CTAATGACCTGTTG TACTTACATTCTC |
| Sequence-based reagent | *Chiloscyllium punctatum Ptch1* forward primer | This paper | PCR primers | TTCAGCCAGA TTGCAGATTACA TCAACC |
| Sequence-based reagent | *Chiloscyllium punctatum Ptch1* reverse primer | This paper | PCR primers | TTCTCTGTGTTTCACA TTCAACGTCCTG |
| Commercial assay or kit | Nextera DNA Sample Preparation Kit | Illumina | Cat# FC-121–1031 | |
| Commercial assay or kit | TruSeq Stranded mRNA LT Sample Prep Kit | Illumina | Cat# RS-122–2101 | |
| Software, algorithm | Trinity | https://github.com/trinityrnaseq/trinityrnaseq | RRID:SCR_013048 | N/A |
| Software, algorithm | Bowtie2 | http://bowtie-bio.sourceforge.net/bowtie2/index.shtml | RRID:SCR_016368 | N/A |
| Software, algorithm | BWA | http://bio-bwa.sourceforge.net/ | RRID:SCR_010910 | N/A |
| Software, algorithm | MACS2 | https://github.com/macs3-project/MACS | RRID:SCR_013291 | N/A |
| Software, algorithm | HOMER | http://homer.ucsd.edu/homer/motif/ | RRID:SCR_010881 | N/A |
| Software, algorithm | RSEM | https://github.com/deweylab/RSEM | RRID:SCR_013027 | N/A |
| Software, algorithm | scikit-learn | https://scikit-learn.org/stable/ | RRID:SCR_002577 | N/A |

## Animals

Animal experiments were conducted in accordance with the guidelines approved by the Institutional Animal Care and Use Committee (IACUC), RIKEN Kobe Branch, and experiments involving mice were approved by IACUC (K2017-ER032). The eggs of brownbanded bamboo shark (*C. punctatum*) were kindly provided by Osaka Aquarium Kaiyukan and were incubated at 25℃ in artificial seawater (MARINE ART Hi, Tomita Pharmaceutical Co., Ltd.) and staged according to the published staging table (*Onimaru et al., 2018*). For mouse embryos, C52BL/6 timed-pregnant females were supplied by the animal facility of Kobe RIKEN, LARGE and sacrificed at different days after 9.5–12.5 days of gestation. For RNA-seq, fin buds and limb buds were dissected in cold seawater and phosphate-buffered saline (PBS), respectively, and stored at −80℃. For in situ hybridization, embryos were fixed overnight in 4% paraformaldehyde in PBS, dehydrated in a graded methanol series, and stored in 100% methanol at −20℃.

## RNA-seq

We sampled mouse forelimb buds at E9.5, E10.5, E11.5 and E12.5 and bamboo shark pectoral fin buds at st27, st27.5, st29, st30, st31, and st32 and pooled several individual samples by stage to obtain enough RNA for each time point. We considered this pooled sample to represent one biological replicate (other replicates were generated using different individuals). Total RNAs from these samples were extracted with the RNeasy Micro and Mini plus kit (QIAGEN, Cat. No. 74034 and 74134) and PicoPure RNA Isolation Kit (ThermoFisher, Cat. No. KIT0214). Genomic DNA was removed with gDNA Eliminator columns included with this kit. For quality control, the Agilent 2100 Bioanalyzer system and Agilent RNA 6000 Nano Kit (Agilent, Cat. No. 5067–1511) were used to

measure the RNA integrity number for each sample. Using 237 ng of each of the RNA samples, strand-specific single-end RNA-seq libraries were prepared with the TruSeq Stranded mRNA LT Sample Prep Kit (Illumina, Cat. No. RS-122–2101 and/or RS-122–2102). For purification, we applied 1.8× (after end repair) and 1.0× (after adapter ligation and PCR) volumes of Agencourt AMPure XP (Beckman Coulter, Cat. No. A63880). The optimal number of PCR cycles for library amplification was determined by a preliminary quantitative PCR using KAPA HiFi HotStart Real-Time Library Amplification Kit (KAPA, Cat. No. KK2702) and was estimated to be 11 cycles for mouse limb buds and 10 cycles for bamboo shark fin buds. The quality of the libraries was checked by Agilent 4200 TapeStation High Sensitivity D1000. The libraries were sequenced after on-board cluster generation for 80 cycles using 1× HiSeq Rapid SBS Kit v2 (Illumina, Cat. No. FC-402–4022) and HiSeq SR Rapid Cluster Kit v2 (Illumina, Cat. No. GD-402–4002) on a HiSeq 1500 (Illumina) operated by HiSeq Control Software v2.2.58 (Run type: SR80 bp). The output was processed with Illumina RTA v1.18.64 for base-calling and with bcl2fastq v1.8.4 for de-multiplexing. Quality control of the obtained fastq files for individual libraries was performed with FASTQC v0.11.5. RNA-seq was performed with three biological replicates for each stage.

## Transcriptome assembly and orthology assignment

We used the NCBI RefSeq mouse proteins (GRCm38.p5; only curated proteins were used) and two bamboo shark gene lists: a genome sequence-based gene model (*Hara et al., 2018*) and transcripts assembled from RNA-seq in this study (see below) for orthology assignment. The amino acid sequences of the published gene model of the bamboo shark are available from https://doi.org/10.6084/m9.fig (*Supplementary file 1*). For the transcriptome assembly, the short reads from the bamboo shark RNA-seq data were trimmed and filtered with Trim Galore! (https://www.bioinformatics.babraham.ac.uk/projects/trim_galore/) and assembled using Trinity v2.4.0 (*Grabherr et al., 2011*; options: –SS_lib_type RF –normalize_max_read_cov 200 min_kmer_cov 2). Protein coding sequences were predicted with a program that finds coding regions, TransDecoder v3.0.1 (*Haas et al., 2013*), according to the guide in TransDecoder (*Supplementary file 2* and *3*). Using these coding gene lists as queries, orthologous pairs were assigned as illustrated in *Figure 1—figure supplement 1*. The idea behind this algorithm is the 'gar bridge' (*Braasch et al., 2016*), an empirical observation that a comparison including intermediate and slowly evolving animals yields a better resolution for identifying homologous sequences than a direct comparison between two species. First, BLASTP v2.7.1 was performed between mouse and bamboo shark genes reciprocally, and also against the coding genes of the elephantfish (or elephant shark; Callorhinchus_milii-6.1.3), spotted gar (LepOcu1), coelacanth (LatCha1), chicken (GRCg6a), alligator (ASM28112v4), and human (GRCh38.p12; options: -outfmt 6 -evalue 1e-30 -window 0). Then, the BLASTP results of bamboo shark queries against the animals listed above (except for the elephantfish) were concatenated, and the best hit across species (cross-species best hit) was identified for each of the bamboo shark genes. If there was no cross-species best hit, then the best hit among the elephantfish genes was retrieved, which may include cartilaginous fish-specific genes. Subsequently, orthologous pairs between mouse and bamboo shark genes were assigned by checking if a cross-species best hit from the bamboo shark BLASTP results also had a best hit in the BLASTP result of mouse genes against the corresponding animal (species-wise best hit; *Supplementary files 4*, *5*, *6*).

For quality control, the orthology of Fgf family members was independently determined by generating molecular phylogenetic trees (*Figure 1—figure supplements 2* and *3*). Amino acid sequences were aligned with an alignment tool, MAFFT v7.419–1 (*Katoh et al., 2002*; options: –localpair –maxiterate 1000) and trimmed with trimAL v1.2 (*Capella-Gutiérrez et al., 2009*; options: -gt 0.9 -cons 60). Then, maximum-likelihood trees were constructed with RaxML v8.2.12 (*Stamatakis, 2014*; options: -x 12345 p 12345 m PROTGAMMAWAG -f a -# 100). The orthology of *Hox* genes was confirmed based on genome synteny. These independently confirmed orthologous pairs were compared with the results of the above orthology assignment algorithm. For a comparison, we also used the results from a reciprocal best hit algorithm, proteinOrtho v6.0.4 (*Lechner et al., 2011*) and the previously generated orthology groups (*Hara et al., 2018*; *Figure 1B*).

## Quantification and scaling

The trimmed RNA-seq short reads were aligned to the transcript contigs for the bamboo shark and curated RefSeq genes (GRCm38.p5) for the mouse using RSEM v1.3.0 (*Li and Dewey, 2011*) and Perl scripts (align_and_estimate_abundance.pl and abundance_estimates_to_matrix.pl) in the Trinity package. TPM (transcripts per million), but not TMM (trimmed mean of M-values), was used for all analyses, because we found some artificial biases in TMM values (see *Figure 1—figure supplement 4*). TPM values from the splicing variants of a single gene were summed up to generate a single value per gene. Then, the means and standard errors of TPM values from three replicates were used for the downstream analyses. Genes with a maximum TPM <1.0 were considered not expressed. For clustering and distance measures, TPM values were scaled so that the maximum value of each gene of each species was set to '1' (Max 1). Whereas this scaling method loses information with respect to the absolute value of the TPMs, it has a substantial advantage when comparisons are being made between evolutionarily distant species. Indeed, previous comparative transcriptome studies have scaled gene expression values in different ways. Among those approaches, the use of Z-scores (standardization) and log transformations are relatively common strategies (e.g. *Kalinka et al., 2010*; *Leiboff and Hake, 2019*; *Levin et al., 2016*). Some researchers have used the intact RPKM (reads per kilobase per million) values to compare closely related species (*Wang et al., 2013*), but, because the RPKM is known to be inconsistent between samples even within a species (*Wagner et al., 2012*). Scaled transcriptional values are commonly used for clustering analyses and visualization of transcriptomic data from different samples within a single species. In this case, scaling is mainly aimed at flattening the dynamic range of transcription levels among genes. For inter-specific comparisons, scaling is also useful for being simultaneously sensitive to differentially regulated genes and also insensitive to conserved housekeeping genes. Here, we examine the effect of several scaling methods and the use of intact TPM values. We define the four relevant scaling methods as follows:

$$M_{g,s,t} = \frac{x_{g,s,t}}{max\left(\{x_{g,s,t}: t = 1..T_s\}\right)}$$

$$Z_{g,s,t} = \frac{\left(x_{g,s,t} - \bar{x}_{g,s}\right)}{\sigma_{g,s}}$$

$$U_{g,s,t} = \frac{x_{g,s,t}}{\left\|\{x_{g,s,t}: t = 1..T_s\}\right\|}$$

$$L_{g,s,t} = log_{10}\left(x_{g,s,t} + 1\right)$$

where $x_{g,s,t}$ is the intact TPM of gene $g$, species $s$, and time point $t$; $T_s$ is the total number of time points in species $s$; $M_{g,s,t}$, $Z_{g,s,t}$, $U_{g,s,t}$ and $L_{g,s,t}$ are scaled values that we refer to as the Max 1, Z-score, Unit vector and Log10 methods, respectively; and $\bar{x}_{g,s}$g,s,1 and $\sigma_{g,s}$g,s,1 are the mean and standard deviation, respectively, of $\{x_{g,s,1}...x_{g,s,T_s}\}$.

First, we take a simple example to develop some intuition as to how these calculations transform TPM values. Let us assume that we compare two species [(species 1 and species 2)], and each species has two genes (gene 1 and gene 2) and three developmental time points (t1, t2, and t3; *Figure 1—figure supplement 5A*). Gene 1 is a constitutively active gene (i.e. a housekeeping gene), and gene 2 is differentially regulated between species. In this example, we want to identify t2 as the most conserved time point because gene two is expressed in both species at this time point. In addition, we want to ignore the subtle expression differences of gene one within and between species. As seen in *Figure 1—figure supplement 5A*, scaling by the Max 1, Unit vector, and Log10 methods effectively conserves the expression dynamics of gene two while suppressing the expression noise of gene 1. In contrast, Z-score scaling amplifies the expression dynamics of both genes to the same degree, which suggests that the Z-score method is sensitive to noise. Calculation of the Euclidean distances for each time point between species 1 and 2 ('Distance' in *Figure 1—figure supplement 5A*) shows that although all scaling methods and the use of intact TPMs indicate that t2 is the most similar time point, Max one creates a greater contrast between conserved and non-conserved time

points than the other methods. Therefore, Max one is likely to be able to sensitively detect inter-specific differences. We also examined a subset of our real transcriptomic data from mouse limb buds and bamboo shark fin buds. As an example, we chose three housekeeping genes conserved in most vertebrates, *Psmd5*, *Mrpl21*, and *Polr1b*—these genes are listed both in a housekeeping gene list https://www.tau.ac.il/~elieis/HKG/HK_genes.txt (*Eisenberg and Levanon, 2013*) and in the BUSCO data set, a gene list used to assess the completeness of genome assemblies (*Simão et al., 2015*). As shown in *Figure 1—figure supplement 5B and C*, the TPM values of these genes were stable throughout developmental time in both species, suggesting that these genes also play a role in the maintenance of basic cellular function in bamboo shark fin development. However, the TPM values of *Mrpl21* and *Polr1b* in mouse limb buds were roughly twice as high as those in bamboo shark fin buds. One explanation for this finding is that the expression of housekeeping genes is low in the bamboo shark because the relatively low temperature of the environment in which it lives slows its metabolic activity. We note, however, that there are many technical uncertainties when directly interpreting TPM values, particularly when comparing distantly related species. For example, differences in DNA sequences of transcripts (such as variations in GC content) between species probably affects the efficiency of library preparation and sequencing. The TPM values are also likely to be biased because of the incompleteness of the reference transcriptome sequence that we used for the bamboo shark (e.g. some genes lack 3' untranlated regions). Therefore, the dynamics of TPM values extracted by scaling methods rather than absolute TPM values are likely to contain more biologically relevant information. Of the scaling methods, Max 1, Unit vector, and Log10 conserved the stable expression profile of the housekeeping genes, whereas the Z-score method amplified the subtle variation in TPM values as seen in the above simple example (*Figure 1—figure supplement 5B*). In particular, the Max one and Unit vector methods transformed the TPM values into relatively comparable values between the two species (compare *Figure 1—figure supplement 5B* with C). For a comparison, we also examined three genes that are heterochronically regulated between bamboo shark fin buds and mouse limb buds (*Figure 1—figure supplement 6A and B*). In this case, all of the scaling methods seemed to conserve the temporal dynamics of gene expression.

To obtain an objective measure, we calculated the ratio of the interspecific Euclidean distance of the three housekeeping genes to that of the three heterochronic genes with different scaling methods (*Figure 1—figure supplement 6C and D*). Namely, the Euclidean distance of expression values was close to zero if we used only housekeeping genes (left panel of *Figure 1—figure supplement 6C*), but it was larger when comparing heterochronic genes (right panel of *Figure 1—figure supplement 6C*). As a result, the Max1 method resulted in the highest ratio (*Figure 1—figure supplement 6D*), suggesting that Max1 is most sensitive to interspecific differences in dynamically regulated genes.

## Clustering analyses of transcriptome data

The scaled values of each orthologous pair were concatenated as a 10-dimensional vector (consisting of four stages for mouse limb buds and six stages for bamboo shark fin buds), and all gene expression vectors were dimensionally reduced with UMAP (hyper parameters: a = 10, b = 1.8) followed by hierarchical clustering (hyper parameters: method = 'ward', metric = 'euclidean'; the code is available at https://github.com/koonimaru/easy_heatmapper; copy archived at https://archive.software heritage.org/swh:1:dir:b1b8edece650ac9e8a7458354aaf69e74f437092;origin=https://github.com/koonimaru/easy_heatmapper;visit=swh:1:snp:a69e903d0efcde99cb203ec86832c5e5c56a43e5;anchor=swh:1:rev:ba1fde133621a52390b82b4c9f73711a56f252b8/). To find genes that have an opposite trend in their expression relative to 'Heterochronic2', a Pearson correlation coefficient (PCC) for TPM values and developmental stages was calculated for each gene for each species, and genes with PCC > 0.5 for bamboo shark fin buds and PCC < −0.5 for mouse limb buds were listed (*Figure 2—figure supplement 1B* and *Supplementary file 8*). For the distance measurements, four different distance methods were calculated: Euclidean distance ($\sqrt{\sum(u_i - v_i)^2}$), correlation distance ($1 - \frac{(u-\bar{u})(v-\bar{v})}{\|(u-\bar{u})\|_2\|(v-\bar{v})\|_2}$), Shannon distance ($-\frac{1}{2}\sum u_i log \frac{(u_i+v_i)}{2u_i} + v_i log \frac{(u_i+v_i)}{2v_i}$), standardized Euclidean distance ($\sqrt{\sum(u_i - v_i)^2/V_i}$), where $u$ and $v$ are gene expression vectors of two samples and $V_i$ is the variance computed over all the values of gene $i$. For PCA analysis, we used the PCA module in a python package, scikit-learn (https://scikit-learn.org/stable/).

For the stage-associated gene analysis in **Figure 3—figure supplement 1B and C**, we first calculated the z-score of each gene at each stage as $\frac{(u_{k,i} - \bar{u}_i)}{\sigma_i}$, where $u_{k,i}$ is the TPM value of gene $i$ at stage $k$, $\bar{u}_i$ is a mean of TPM over all the stages, and $\sigma_i$ is the standard deviation of the TPM. Genes with TPM $\geq 1.0$ and the absolute Z-score $\geq 1.0$ were counted as stage-associated genes. For the tissue-associated gene analysis, the entropy of each gene was calculated using RNA-seq data of 71 tissues downloaded from the ENCODE web site (https://www.encodeproject.org/; see **Supplementary file 10** for all list). Entropy was calculated as follows:

$$p_{k,i} = \frac{TPM_{k,i}}{\sum_k TPM_{k,i}}$$

$$H_i = -\sum_k p_{k,i} log(p_{k,i})$$

where $TPM_{k,i}$ is the TPM value of gene $i$ in tissue $k$, $p_{k,i}$ is a probability distribution and $H_i$ is entropy. Genes with TPM (of mouse limb buds) $\geq 1.0$ and $0.65 \leq$ entropy were counted as tissue-associated genes.

## Whole-mount in situ hybridization

To clone DNA sequences for RNA probes, we used primers that were based on the nucleotide sequences in the ENSEMBL database (https://www.ensembl.org) for mouse genes and in the transcriptome assembly (**Supplementary file 3**); bamboo shark *Hand2* (Chipun0004250/g4250.t1/ TRINITY_DN85524_c0_g1_i1), 5′-ACCAGCTACATTGCCTACCTCATGGAC-3′ and 5′-CACTTG TTGAACGGAAGTGCACAAGTC-3′; bamboo shark *Vcan* (Chipun0003941/g3941.t1/ TRINITY_DN95522_c0_g1_i8), 5′-AGCTTGGGAAGATGCAGAGAAGGAATG-3′ and 5′-AGAGCAGCTTCA-CAATGCAGTCTCTGG-3′; bamboo shark *Hoxd12* (Chipun0005654/g5654.t1/ TRINITY_DN85970_c0_g1_i1), 5′-GCCAGTATGCAACAGATCCTCTGATGG-3′ and 5′-CTAATGACC TGTTGTACTTACATTCTC-3′; bamboo shark *Ptch1* (Chipun0003320/g3320.t1/TRINITY_DN92499_c0_g1_i3), 5′-TTCAGCCAGATTGCAGATTACATCAACC-3′ and 5′-TTCTCTGTG TTTCACATTCAACGTCCTG-3′; bamboo shark *Aldh1a2* (Chipun0010503/g10503.t1/TRINITY_DN81423_c0_g1_i1), 5′-TTGAACTTGTACTAAGTGGTATCGCTG-3′ and 5′-AGGATGTGAACA TTAGGCTGACCTCAC-3′; mouse *Hand2* (ENSMUST00000040104.4), 5′-ACCAAACTCTCCAAGA TCAAGACACTG-3′ and 5′-TTGAATACTTACAATGTTTACACCTTC-3′; mouse *Vcan* (ENSMUST00000109546.8), 5′-TGCAAAGATGGTTTCATTCAGCGACAC-3′ and 5′-ACACGTGCAGA-GACCTGCAAGATGCTG-3′; mouse *Hoxd12* (ENSMUST00000109546.8), 5′-TGCAAAGATGGTTTCA TTCAGCGACAC-3′ and 5′-ACACGTGCAGAGACCTGCAAGATGCTG-3′; mouse *Aldh1a2* (ENSMUST00000034723.5), 5′-ACCGTGTTCTCCAACGTCACTGATGAC-3′ and 5′-TCTGTCAGTAA-CAGTATGGAGAGCTTG-3′; mouse *Ptch1* (ENSMUST00000192155.5), 5′-GGGAAGGCAGTTCATTG TTACTGTAACTG-3′ and 5′-TGTAATACGACTCACTATAGGTCAGAAGCTGCCACACACAGGCA TGAAGC-3′. Note that although we also tried bamboo shark *Shh* expression analysis using several RNA probes, we did not obtain specific signals. Fixed embryos were processed for in situ hybridization as described (**Westerfield, 2000**) with slight modifications. Briefly, embryos were re-hydrated with 50% MeOH in PBST (0.01% Tween 20 in PBS) and with PBST for 5–30 min each at room temperature (RT). Then, embryos were treated with 20 µg/ml proteinase K (Roche) in PBST (5 s for mouse E11.5 and E12.5 embryos, 5 min for st. 27 and st. 29 bamboo shark embryos, 10 min for st. 31 and st. 32 bamboo shark embryos). After the proteinase treatment, embryos were fixed in 4% paraformaldehyde/PBS for 1 hr, followed by one or two washes with PBST for 5–10 min each. Optionally, if embryos had some pigmentation, they were immersed in 2% $H_2O_2$ until they became white. Then, embryos were incubated for 1 hr in preheated hybridization buffer (50 ml formaldehyde; 25 ml 20× SSC, pH 5.0; 100 µl 50 mg/ml yeast torula RNA; 100 µl 50 mg/ml heparin; 1 ml 0.5 M EDTA; 2.5 ml 10% Tween 20; 5 g dextran sulfate; and DEPC-treated MilliQ water to a final volume of 100 ml) at 68°C. Subsequently, embryos were incubated with fresh hybridization buffer containing 0.25–4 µl/ml of RNA probes at 68°C overnight. Embryos were washed twice with preheated Wash buffer 1 (50 ml formaldehyde; 25 ml 20× SSC, pH 5.0; 2.5 ml 10% Tween 20; and DEPC-treated MilliQ water to a final volume of 100 ml) for 1 hr each at 68°C; once with preheated Wash buffer 2, which consisted of equal volumes of Wash buffer 1 and 2× SSCT (10 ml 20× SSC, pH 7.0; 1 ml 10% Tween 20; and

MilliQ water to a final volume of 100 ml), for 10 min at 68°C; once with preheated 2× SSCT at 68°C for 10 min; and once with TBST at room temperature for 10 min. Embryos were then incubated with a blocking buffer (20 µl/ml 10% bovine serum albumin, 20 µl/ml heat-inactivated fetal bovine serum in TBST) for 1 hr at room temperature, followed by incubation with 1/4000 anti-digoxigenin (Roche) in fresh blocking buffer at 4°C overnight. Embryos were washed four times with TBST for 10–20 min each and were incubated at 4°C overnight. Finally, embryos were incubated with NTMT (200 µl 5 M NaCl; 1 ml 1 M Tris-HCl, pH 9.8; 500 µl 1 M MgCl$_2$; 100 µl 10% Tween 20; and MilliQ water to a final volume of 10 ml) for 20 min and then with 15 µg/ml nitro-blue tetrazolium chloride (NBT) and 175 µg/ml 5-bromo-4-chloro-3-indolyphosphate *p*-toluidine salt (BCIP) in NTMT for 10 min to 2 hr until signals appeared. Pictures were taken with an Olympus microscope. For bamboo shark embryos, experiments were performed for at least two biological replicates.

## ATAC-seq

Mouse forelimb buds at E9.5, E10.5, E11.5, and E12.5 were dissected, and samples from several individuals were pooled by stage to obtain enough cells. We considered this pooled sample to represent a biological replicate (other replicates were generated using different individuals). To obtain single-cell suspensions, pooled samples were treated with collagenase for 10 min at room temperature. The tissues were then dissociated into single-cell suspensions by pipetting the mixture and passing it through a 40 µm mesh filter (Funakoshi, Cat. No. HT-AMS-14002); the cell suspension was frozen in CryoStor medium (STEMCELL Technologies, Cat. No. ST07930) with Mr. Frosty (Thermo Scientific, Cat. No. 5100–0001) at −80°C overnight, according to *Milani et al., 2016*. An ATAC-seq library was prepared as described (*Buenrostro et al., 2013*) with some minor modifications. For library preparation, stored cells were thawed in a 38°C water bath and centrifuged at 500 g for 5 min at 4°C, which was followed by a wash using 50 µl of cold PBS and a second centrifugation at 500 g for 5 min at 4°C. Ten thousand cells per sample were collected, without distinguishing dead cells, and were lysed using 50 µl of cold lysis buffer (10 mM Tris-HCl, pH 7.4; 10 mM NaCl; 3 mM MgCl$_2$; and 0.1% IGEPAL CA-630). Immediately after lysis, cells were spun at 1000 g for 10 min at 4°C, and the supernatant was discarded. For the transposition reaction, cells were re-suspended in the transposase reaction mix (25 µl 2× TD buffer, 2.5 µl Tn5 transposase [in the Nextera DNA Sample Preparation Kit, Illumina, Cat. No. FC-121–1031], and 22.5 µl nuclease-free water) and incubated for 30 min at 37°C. The reaction mix was purified using DNA Clean and Concentrator-5 (Zymo Research, Cat. No. D4014) by adding 350 µl of DNA-binding buffer and eluting in a volume of 10 µl. After a five-cycle pre-PCR amplification, the optimal number of PCR cycles was determined by a preliminary PCR using KAPA HiFi HotStart Real-Time Library Amplification Kit and was estimated to be four cycles. The PCR products were purified using 1.8× volumes of Agencourt AMPure XP. As a control, 50 ng of mouse genomic DNA was also transposed following the standard procedure of the Nextera DNA Sample Preparation Kit. Sequencing with HiSeq X was outsourced to Macrogen, Inc, which was carried out with HiSeq Control Software 3.3.76 (Run type: PE151bp). The output was processed with Illumina RTA 2.7.6 for base-calling and with bcl2fastq 2.15.0 for de-multiplexing. Quality control of the obtained fastq files for individual libraries was performed with FASTQC v0.11.5. ATAC-seq was performed with three biological replicates for each stage.

## ATAC-seq data analysis

The short-read data from ATAC-seq were trimmed and filtered with Trim-Galore! (v0.5.0; options: `--paired --phred33 -e 0.1 -q 30`). We also removed reads that originated from mitochondrial genome contamination by mapping reads to the mouse mitochondrial genome using bowtie2 v2.3.4.1 (*Langmead and Salzberg, 2012*). The rest of the reads were mapped onto the mouse genome (mm10) using bwa v0.7.17 with the 'mem' option (*Li and Durbin, 2010*). Among the mapped reads, we removed reads with length >320 bp to reduce noise. The rest of the reads were further down-sampled to around 83.2 million reads to equalize the sequence depth of every sample. Peak calls were done with MACS2 v2.1.1 (*Zhang et al., 2008*; options: `--nomodel --shift −100 --extsize 200 f BAMPE -g mm -B -q 0.01`; the genomic reads were used as a control for all samples). For FRiP score calculation, a module, 'countReadsPerBin.CountReadsPerBin' in deepTools v3.2.1 (*Ramírez et al., 2016*), was used to count reads in peaks, and these read counts were then divided by the total number of reads. To evaluate reproducibility among the replicates, we first divided the

mouse genome into 500 bp bins. Then, the ATAC-seq peaks were re-distributed into these bins with bedtools (*Quinlan and Hall, 2010*; options: intersect -F 0.4 f 0.4 -e -wo). Peaks of >500 bp were subdivided into 500-bp-long regions, and those of <500 bp were extended to fit within the closest 500 bp window. Subsequently, these peaks were converted into one-hot vectors, in which '1' means that a 500-bp-long genomic region harbors an ATAC-seq peak. Genomic regions that lacked ATAC-seq peaks in all data were omitted. Using these one-hot vectors, Euclidean distances between the ATAC-seq data were calculated (*Figure 5—figure supplement 1A*).

For the conservation analysis, the significant variation in the length of ATAC-seq peaks complicated this evaluation. To deal with such variation, we the ATAC-seq peaks were re-distributed into 100 bp bins with bedtools (*Quinlan and Hall, 2010*; options: intersect -F 0.4 f 0.4 -e -wo) as described above. The sequences in these peaks were retrieved with BLASTN v2.7.1 against the genomes of 16 vertebrate species listed in *Supplementary file 10* (BLASTN options: -task dc-megablast -max_target_seqs 1). The blast hits that scored $\geq$40 were considered as conserved sequences. In this way, the final figures shown in *Figure 5C* represent the fraction of the total conserved sequence length in the peaks of each stage rather than the number of conserved peaks. For confirmation, we also used a different alignment algorithm, LAST v961 (*Kiełbasa et al., 2011*) to find conserved sequences. To generate mouse genome databases for LAST, we first masked repeat sequences with N and split the genome file into multiple files, each of which contained a single chromosome sequence. Then, databases were generated using lastdb (options: -cR01). Alignments with the bamboo shark genome (Cpunctatum_v1.0; https://transcriptome.riken.jp/squalomix/resources/01.GCA_003427335.1_Cpunctatum_v1.0_genomic.rn.fna.gz) and the alligator genome (ASM28112v3) were carried out by lastal (options: -a1 -m100). Only a unique best alignment was selected using last-split. These alignment results were converted into the bed format, and regions that overlapped with the ATAC-seq peaks that were subdivided into 100 bp bins were counted.

For the clustering analysis, we converted the alignment files of the ATAC-seq reads into mapped reads in bins per million (BPM) coverage values with 200 bp resolution using bamCoverage in deepTools v3.2.1 (*Ramírez et al., 2016*; options: -of bedgraph –normalizeUsing BPM –effective-GenomeSize 2652783500 -e -bs 200). Then, BPMs at the summits of ATAC-seq peaks and an additional 600 bp to the left and to the right of each summit (1400 bp in total) were collected and clustered by t-SNE (https://github.com/DmitryUlyanov/Multicore-TSNE; hyper parameters: perplexity = 30.0, n_iter = 5000) followed by hierarchical clustering (hyper parameters: method = 'ward', metric = 'euclidean'). Enriched motifs were detected using a Perl script, findMotifsGenome.pl in HOMER v4.10.4 (*Heinz et al., 2010*; options: -size 100 -mask). To count the number of motif occurrences, '-find' option of findMotifsGenome.pl was used, and sequences that scored $\geq$75% of the highest motif score were counted. For GO analysis, annotatePeaks.pl in HOMER was used. For the tissue-specificity analysis, we downloaded several aligned and unaligned reads of ATAC-seq experiments on 25 different tissues from the ENCODE web site (https://www.encodeproject.org/; see *Supplementary file 10* for a complete list), and peaks were called as described above. Then, peaks that did not overlap with other tissues/cells were detected using bedtools.

## Data and materials availability

RNA-seq and ATAC-seq data sets generated during the current study are available in the Gene Expression Omnibus (GEO) repository under accession number GSE136445. Other sequence data and raw data are available in the figshare (DOI: 10.6084/m9.figshare.9928541). Code for clustering analysis is available at https://github.com/koonimaru/easy_heatmapper. Materials related to this paper are available upon request from the corresponding authors.

## Acknowledgements

We thank Kenta Sumiyama, James Shape and Masahiro Uesaka for fruitful discussions; Itoshi Nikaido and Laboratory for Bioinformatics Research for providing computational resources; Itsuki Kiyatake, Kiyonori Nishida, and Osaka Aquarium Kaiyukan for kindly providing bamboo shark eggs; Laboratory for Animal Resources and Genetic Engineering for supplying mouse embryos; the ENCODE consortium and the ENCODE production laboratories for generating ATAC-seq data sets.

## Additional information

### Funding

| Funder | Grant reference number | Author |
|---|---|---|
| Japan Society for the Promotion of Science | 17K15132 | Koh Onimaru |
| Ministry of Education, Culture, Sports, Science and Technology | | Koh Onimaru<br>Kaori Tatsumi<br>Chiharu Tanegashima<br>Mitsutaka Kadota<br>Osamu Nishimura<br>Shigehiro Kuraku |
| RIKEN | | Koh Onimaru |

The funders had no role in study design, data collection and interpretation, or the decision to submit the work for publication.

### Author contributions

Koh Onimaru, Conceptualization, Resources, Data curation, Supervision, Funding acquisition, Validation, Investigation, Visualization, Methodology, Writing - original draft, Project administration; Kaori Tatsumi, Chiharu Tanegashima, Mitsutaka Kadota, Osamu Nishimura, Data curation, Validation, Methodology, Writing - review and editing; Shigehiro Kuraku, Conceptualization, Supervision, Funding acquisition, Validation, Project administration, Writing - review and editing

### Author ORCIDs

Koh Onimaru (iD) https://orcid.org/0000-0002-2428-9510
Osamu Nishimura (iD) http://orcid.org/0000-0003-1969-2580
Shigehiro Kuraku (iD) https://orcid.org/0000-0003-1464-8388

### Ethics

Animal experimentation: Animal experiments were conducted in accordance with the guidelines approved by the Institutional Animal Care and Use Committee (IACUC), RIKEN Kobe Branch, and experiments involving mice were approved by IACUC (K2017-ER032).

### Decision letter and Author response

Decision letter https://doi.org/10.7554/eLife.62865.sa1
Author response https://doi.org/10.7554/eLife.62865.sa2

## Additional files

### Supplementary files

• Supplementary file 1. Summary of short-read sequencing data.

• Supplementary file 2. Orthology asignment for the transcriptome of the brown-banded bamboo shark. Columns 1–4: transcriptome assembly ID, NCBI gene ID, gene symbol, blast score.

• Supplementary file 3. Orthology asignment for the gene model of the brown-banded bamboo shark. Columns 1–4: gene model ID, NCBI gene ID, gene symbol, blast score.

• Supplementary file 4. Quality control of orthology assignment. Source data to create *Figure 1C*.

• Supplementary file 5. The mean and SEM of TPM values of mouse limb RNA-seq data. Source data for *Figure 1D* and other plots related to gene expression amount.

• Supplementary file 6. The mean and SEM of TPM values of bamboo shark fin RNA-seq data. Source data for *Figure 1D* and other plots related to gene expression amount.

• Supplementary file 7. Clustered gene expression table with phenotype annotation. The details of *Figure 2A*.

• Supplementary file 8. The list of genes downregulated over time in mouse limb buds being upregulated in bamboo shark fin buds over time (related to *Figure 2—figure supplement 1B*).

• Supplementary file 9. PCA loadings of *Figure 4D and E*.

• Supplementary file 10. List of public data used in *Figures 4*, *5* and *6*.

• Supplementary file 11. GO analysis of ATAC-seq peaks. c1 to c8 correspond to the clusters in *Figure 5A*.

• Transparent reporting form

## Data availability

RNA-seq and ATAC-seq datasets generated during the current study are available in the Gene Expression Omnibus (GEO) repository under accession number GSE136445. Data necessary to reproduce this study are deposited in figshare (https://figshare.com/articles/Onimaru_et_al_Supplementary_Data/9928541; https://doi.org/ 10.6084/m9.figshare.9928541). Code for clustering analysis is available at https://github.com/koonimaru/easy_heatmapper copy archived at https://archive.softwareheritage.org/swh:1:rev:ba1fde133621a52390b82b4c9f73711a56f252b8/. The following previously published datasets from ENCODE were used: ENCFF478FHR ENCFF955MIX ENCFF210MKE ENCFF431KXE ENCFF275OKU ENCFF426VDN ENCFF002LRT ENCFF576SKK ENCFF592ZRO ENCFF798QON ENCFF336VLY ENCFF407NCE ENCFF572CMB ENCFF695FLH ENCFF130WMA ENCFF224JRS ENCFF337ETV ENCFF535DAV ENCFF540ZEZ ENCFF762LUG ENCFF279LMU ENCFF518FYP ENCFF377YCK ENCFF086MTT ENCFF064NKM ENCFF406EUS ENCFF258GFE ENCFF031SEH ENCFF694SPD ENCFF051GLX ENCFF304CCF ENCFF655OFT ENCFF483MKX ENCFF007HEF ENCFF550NVA ENCFF848NLJ ENCFF929LOH ENCFF382CMV ENCFF360MVK ENCFF159HYY ENCFF618OJP ENCFF329VCX ENCFF341HRL ENCFF894ZND ENCFF702NAP ENCFF109LQF ENCFF146ZCO ENCFF154RTC ENCFF709ZKC ENCFF040SPZ ENCFF912PDM ENCFF141JSP ENCFF985YPA ENCFF064JRU ENCFF774MTJ ENCFF376TIM ENCFF612QXM ENCFF584HRP ENCFF353TSI ENCFF583FIG ENCFF143XEE ENCFF590KVK ENCFF107GOQ ENCFF370RSB ENCFF906UHI ENCFF034BFB ENCFF928FUL ENCFF872PTK ENCFF982ZSW ENCFF454BSG ENCFF035UJZ ENCFF471VWH ENCFF501QKH ENCFF113PQF ENCFF322CQL ENCFF622HGW ENCFF746ASR ENCFF232GHI ENCFF484RFZ ENCFF658OKS ENCFF232PNH ENCFF403VCU ENCFF688KUB ENCFF815LLD ENCFF557YZH ENCFF636YTN ENCFF142IPK ENCFF387ORM ENCFF877QHQ ENCFF877LFX ENCFF994LOF ENCFF398KDL ENCFF618YMO ENCFF598ZGD ENCFF924SYL ENCFF809YXL ENCFF685HFN ENCFF697FTK ENCFF887QYY ENCFF171GOW ENCFF790TWE ENCFF635MWR ENCFF818OKO ENCFF978ZGA ENCFF645HNE ENCFF237MEG ENCFF738MPC ENCFF905ZTZ ENCFF914USA ENCFF417HDL ENCFF105XRN ENCFF302YAI ENCFF502HEW ENCFF978POS ENCFF107SIK ENCFF143SWD ENCFF311YQH ENCFF940KCT.

The following datasets were generated:

| Author(s) | Year | Dataset title | Dataset URL | Database and Identifier |
|---|---|---|---|---|
| Onimaru K | 2019 | A comparison of evolutionary changes and constraints on gene regulation between fin and limb development | https://www.ncbi.nlm.nih.gov/geo/query/acc.cgi?acc=GSE136445 | NCBI Gene Expression Omnibus, GSE136445 |
| Onimaru K, Tatsumi K, Tanegashima C, Kadota M, Nishimura O, Kuraku S | 2020 | Onimaru_et_al_Supplementary_Data | https://figshare.com/articles/Onimaru_et_al_Supplementary_Data/9928541 | figshare, 10.6084/m9.figshare.9928541 |

The following previously published datasets were used:

| Author(s) | Year | Dataset title | Dataset URL | Database and Identifier |
|---|---|---|---|---|
| Nantong University | 2015 | Gekko_japonicus_V1.1 | https://www.ncbi.nlm.nih.gov/assembly/GCF_001447785.1/ | NCBI Assembly, GCF_001447785.1 |
| Broad Institute | 2011 | AnoCar2.0 | https://www.ncbi.nlm.nih.gov/assembly/GCF_000090745.1/ | NCBI Assembly, GCF_000090745.1 |
| International Crocodilian Genomes Working Group | 2009 | ASM28112v4 | https://www.ncbi.nlm.nih.gov/assembly/GCF_000281125.3/ | NCBI Assembly, GCF_000281125.3 |
| Genome Reference Consortium | 2018 | GRCg6a | https://www.ncbi.nlm.nih.gov/assembly/GCF_000002315.6 | NCBI Assembly, GCF_000002315.6 |
| Beijing Genomics Institute | 2012 | GeoFor_1.0 | https://www.ncbi.nlm.nih.gov/assembly/GCF_000277835.1/ | NCBI Assembly, GCF_000277835.1 |
| P. sinensis genome project consortium | 2012 | PelSin_1.0 | https://www.ncbi.nlm.nih.gov/assembly/GCF_000230535.1/ | NCBI Assembly, GCF_000230535.1 |
| Painted turtle genome sequencing consortium | 2014 | Chrysemys_picta_bellii-3.0.3 | https://www.ncbi.nlm.nih.gov/assembly/GCF_000241765.3/ | NCBI Assembly, GCF_000241765.3 |
| Max Planck Society | 2018 | ASM291563v1 | https://www.ncbi.nlm.nih.gov/assembly/GCA_002915635.1/ | NCBI Assembly, GCA_002915635.1 |
| DOE Joint Genome Institute | 2016 | Xenopus_tropicalis_v9.1 | https://www.ncbi.nlm.nih.gov/assembly/GCF_000004195.3/ | NCBI Assembly, GCF_000004195.3 |
| Broad Institute | 2011 | LatCha1 | https://www.ncbi.nlm.nih.gov/assembly/GCF_000225785.1/ | NCBI Assembly, GCF_000225785.1 |
| Broad Institute | 2012 | LepOcu1 | https://www.ncbi.nlm.nih.gov/assembly/GCF_000242695.1/ | NCBI Assembly, GCF_000242695.1 |
| Genome Reference Consortium | 2017 | GRCz11 | https://www.ncbi.nlm.nih.gov/assembly/GCF_000002035.6/ | NCBI Assembly, GCF_000002035.6 |
| The University of Tokyo | 2017 | ASM223467v1 | https://www.ncbi.nlm.nih.gov/assembly/GCF_002234675.1/ | NCBI Assembly, GCF_002234675.1 |
| Institute of Molecular and Cell Biology, Singapore | 2013 | Callorhinchus_milii-6.1.3 | https://www.ncbi.nlm.nih.gov/assembly/GCF_000165045.1/ | NCBI Assembly, GCF_000165045.1 |
| Phyloinformatics Unit, Division of Bio-Function Dynamics Imaging, Center for Life Science Technologies, RIKEN | 2018 | Storazame_v1.0 | https://www.ncbi.nlm.nih.gov/genome/12714?genome_assembly_id=397946 | NCBI Assembly, GCA_003427355.1 |
| Phyloinformatics Unit, Division of Bio-Function Dynamics Imaging, Center for Life Science Technologies, RIKEN | 2018 | Cpunctatum_v1.0 | https://www.ncbi.nlm.nih.gov/genome/12366?genome_assembly_id=397945 | NCBI Assembly, GCA_003427335.1 |

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
