## [Decision Letter]

**Acceptance summary:**

This manuscript presents cutting edge data to compare the development of shark fin and mouse limb and an important discovery – the existence of a conserved mid-developmental stage in paired appendage development. This study is timely and important, and will make an excellent publication to *eLife*.

**Decision letter after peer review:**

[Editors’ note: the authors submitted for reconsideration following the decision after peer review. What follows is the decision letter after the first round of review.]

Thank you for submitting your work entitled "Developmental hourglass and heterochronic shifts in fin and limb development" for consideration by *eLife*. Your article has been reviewed by two peer reviewers, and the evaluation has been overseen by a Reviewing Editor and a Senior Editor. The following individual involved in review of your submission has agreed to reveal their identity: Jose Luis Gomez-Skarmeta (Reviewer #2).

Our decision has been reached after consultation between the reviewers. Based on these discussions and the individual reviews below, we regret to inform you that your work will not be considered further for publication in *eLife*.

While the general scope of the paper is potentially suitable for *eLife*, it requires substantial additional analyses and discussion, as outlined in the reviews. In particular, it would be important to show the pattern of expression of Shh and its main targets in the bamboo shark to sustain the conclusions in Figure 2. Given that these experiments would likely take longer than the normal revision times for *eLife* papers, we reject the current version for now, but would consider a substantially revised new version.

Reviewer #1:

In this manuscript, to identify differences between limbs and fins, the authors generate and compare the temporal transcriptomes of mouse forelimb and bamboo shark pectoral fins. The comparison reveals a notable heterochrony of gene expression between limbs and fins. The analysis of distances of transcriptome profiles indicate stronger conservation at intermediate stages (hourglass-shaped conservation). Next the authors generate the ATAC-seq profiles of developing limb buds and find that conserved regulatory sequences are most active during mid-stage limb development.

This is an interesting study that requires some additional analysis and discussion to better sustain the conclusions reached.

Major concerns:

A major concern regarding this work is the use of the whole limb/fin for the transcriptomic and ATAC analyses. The limb bud is very heterogenous, even more as it develops, and this makes it difficult to extract conclusions using this kind of bulk analysis, more taking into account that late processes such as chondrogenic differentiation may greatly vary between mouse and shark.

Another concern is the scaling (maximum TPM=1, minimum TPM=0). While I agree that this helps capturing the dynamics of gene expression, it does not reflect the magnitude of the change and could lead to misleading interpretations. I think this may happen with the interpretation of the Shh pathway

1) Figure 2A: More information on the list of genes in each of the categories after gene-by-gene comparison of expression dynamics should be provided and discussed rather than only mentioning a couple of genes. The list should also be provided (i.e. excel). Particularly interesting is the inverse behavior of some genes in the "Heterochrony" group that are downregulated over time in the mouse limb bud being upregulated in the shark fin.

2) Figure 2D-E- I think that some hybridizations for Shh and its main targets in the bamboo shark are required to sustain the conclusions from these two panels. It may be that the sustained expression in the mouse corresponds to later chondrogenic stages that have already started at E12.5 and the whole heterochrony responding to different time resolution as mentioned by the authors. I don't agree with the authors in that the expression dynamics of HoxA/D genes is similar in both species, at least for the 5' members.

3) The consideration of ATAC sequences as active sequences should be softened as this is not always hold true. My interpretation is that most changes happen between 9.5 and 10.5, rather than conserved sequences being more active at E10.5

Reviewer #2:

This is an interesting study which aims to identify differences between fins and limbs. Performing transcriptomic comparison between pectoral fins from a non-model Chondricthyan species and forelimbs from mouse across a series of developmental stages, Onimaru et al. show that a noteworthy number of genes shows a heterochronic shift, alias a reverse temporal dynamic of expression between the two species. Moreover, they present an hourglass-shaped conservation of gene expression, but also of active regulatory regions in middle stages of development. Interestingly, in these stages they also detect more tissue- and stage-specific enhancers leading to the hypothesis that the middle developmental stages are evolutionary constrained by the increased regulatory complexity over pleiotropic genes.

This work shows how comparing distant species constitutes a good approach to understand how morphological novelties occur or are constrained during evolution and hints towards some of the changes that might have occurred during fin-to-limb transition. The data that Onimaru et al. have produced are also a good resource for the scientific community and the overall work leads to many interesting follow-up questions. Due to all the above reasons, I support the publication of this article in *eLife*.

However, the analyses presented are not always described as clearly or in-depth as desired and few observations are overstated. Therefore, I recommend the implementation of the comments below to strengthen the reliability, to enrich the content of the data presented and to prevent any confusion for the reader.

1) In Figure 2A, using hierarchical clustering the authors show that there is a heterochronic shift in gene expression between mouse limbs and shark fins. However, this group consists of different subclusters which not all follow exactly the general trend that the authors describe in their results. Could the authors discuss on these genes that still show different temporal dynamics of expression between sharks and mouse, but do not show opposite -timewise- trend than in mouse?

2) The authors should provide individual tables for each cluster described in Figure 2A (fin-specific, limb-specific, stable, conserved/late, heterochronic) instead of the Supplementary file 3, which is quite confusing in its current form.

3) In Figure 5, which are the GO terms associated to the genes for these clusters? What are the enriched motifs in cluster 8, largely specific to E9.5, and the GOs of the associated genes? The full list of motifs and associated genes for each cluster should be available. Moreover, is the conservation degree of the ATAC peaks different for each cluster?

4) As far as the HOMER analysis is concerned in Figure 5, why the authors used the extended sequence length of 1400 bp to perform TF motif analysis? Also, did they perform the enrichment analysis with default HOMER options? If so, random genomic regions were used as a statistical background. Can these results be replicated when using a more biologically relevant background? For example, the peaks of Cluster5 are enriched in CTCF when compared to random regions of the DNA (the default HOMER approach), but are they also enriched in CTCF when compared to all the open chromatin regions that were detected during development?

5) Results first paragraph: the figure supplement 4, not 1, refers to the details of RNA-seq data.

6) Have the authors used all three replicates in the transcriptomic analyses? We could assume that due to the last sentence referring to that in Figure 1—figure supplement 4, but it should be clearly stated.

7) Could the authors explain why they used only one of the replicates for the ATAC-seq hierarchical clustering in Figure 5A and comment whether the ATAC-seq peaks tested were present in all 3 replicates?

[Editors’ note: further revisions were suggested prior to acceptance, as described below.]

Thank you for submitting your article "Developmental hourglass and heterochronic shifts in fin and limb development" for consideration by *eLife*. Your article has been reviewed by two peer reviewers, and the evaluation has been overseen by a Reviewing Editor and Detlef Weigel as the Senior Editor. The following individual involved in review of your submission has agreed to reveal their identity: Gunter Wagner (Reviewer #2).

The reviewers have discussed the reviews with one another and the Reviewing Editor has drafted this decision to help you prepare a revised submission.

We would like to draw your attention to changes in our revision policy that we have made in response to COVID-19 (https://elifesciences.org/articles/57162). Specifically, we are asking editors to accept without delay manuscripts, like yours, that they judge can stand as *eLife* papers without additional data, even if they feel that they would make the manuscript stronger. Thus the revisions requested below primarily address clarity and presentation.

Summary:

The authors present a detailed RNAseq and ATACseq comparison of mouse limb development and bamboo shark pectoral fin development. This study provides a wealth of functional genomic data leading to an important discovery: the existence of a mid – developmental "hour glass like" constrained stage of limb development. This is a significant discovery because it suggests a mechanistic basis for conserved developmental identities as the sub-organismal level. The homology of paired fins and limbs is not in question since upwards of 200 years, but understanding the developmental/mechanistic basis for this fact is a still unresolved issue in biology. This paper makes an important step in resolving this issue.

Revisions:

Given the recent publication of Dr. Woltering (https://pubmed.ncbi.nlm.nih.gov/32875118/), the authors may want to comment on this paper in relation with the Shh, Hox expressions they report.

We suggest that the authors indicate, at least in Materials and methods, their failure to detect Shh expression by ISH. Knowing this may be of help for other researchers.

Results paragraph three: Figure 1—figure supplement 7 instead of 8?

Introduction: the expression of Hoxa11 and Hoxa13 is actually not conserved in fin development, because the critical spatial exclusion of their expression domains is NOT seen in fins, even though a distal bias of Hoxa13 expression is shared. Please correct.

It is surprising to find 16,442 orthologs between shark and mouse, given that 1-1 orthologs among a sample of 10 eutherian species finds only <8,000 orthologs. A comment on this finding might be in order.

Results paragraph three: a non-colinear relationship in Hoxd gene expression levels also applies to chicken wings, where Hoxd12 is higher expressed than all the others, but I am not sure that was ever published. This could point to a scenario where mouse limb development is not as paradigmatic as it often seems.

Subsection “Comparison of SHH signaling pathways in limb and fin buds”: it is hard to see how Shh delayed onset can be supported without a rigid mapping of developmental stages between fin and limb development.

Discussion paragraph three: Please add reference to Piasecka et al., 2013.

Discussion paragraph four: We think it is important to be precise here. The correct statement is that mutations affecting this stage have more dramatic fitness consequences, rather than that it is less susceptible to mutation. What creates this impression is that the substitution rate is less not necessarily the mutation rate, as the authors note in the next sentence.

---

## [Author Response]

[Editors’ note: the authors resubmitted a revised version of the paper for consideration. What follows is the authors’ response to the first round of review.]

Reviewer #1:In this manuscript, to identify differences between limbs and fins, the authors generate and compare the temporal transcriptomes of mouse forelimb and bamboo shark pectoral fins. The comparison reveals a notable heterochrony of gene expression between limbs and fins. The analysis of distances of transcriptome profiles indicate stronger conservation at intermediate stages (hourglass-shaped conservation). Next the authors generate the ATAC-seq profiles of developing limb buds and find that conserved regulatory sequences are most active during mid-stage limb development.This is an interesting study that requires some additional analysis and discussion to better sustain the conclusions reached.Major concerns:A major concern regarding this work is the use of the whole limb/fin for the transcriptomic and ATAC analyses. The limb bud is very heterogenous, even more as it develops, and this makes it difficult to extract conclusions using this kind of bulk analysis, more taking into account that late processes such as chondrogenic differentiation may greatly vary between mouse and shark.

We agree that our comparative transcriptomic analysis has certain limitations owing to the use of bulk samples. In particular, whereas our analysis can capture overall temporal expression differences between the fin and limb buds, spatial information is lacking in our data. However, in contrast to the reviewer’s concern, we found that our data correctly identified some differentiation processes such as chondrogenesis and myogenesis (Figure 1—figure supplement 8).

In the case of chondrogenesis, bamboo shark fin buds and mouse limb buds share the early rise of *Sox9*, *Runx3* and *Tgfb2* expressions (key factors for chondrogenic differentiation). Subsequently, a cartilage-specific proteoglycan, *Acan* is upregulated slightly later in both species. On the other hand, *Noggin*, which is used for early chodrogenic marker in mouse limb development, was not detected neither in the genome nor transcriptome assemblies of the bamboo shark. Therefore, while it is convincing that the chondrogenic process of the bamboo shark have varied due to the lack of endochondral ossification in cartilaginous fishes, our trascriptome data revealed a partially conserved process for chondrogenic differentiation.

Regarding myogenesis, our trascriptome data captures a myogenic process conserved in the both species: Pax3 (a marker of myogenic precursor cells) is downregulated over the developmental time, and MyoD gene family (MyoG, MyoD1, Myf5) takes turns for further differentiation. On the other hand, whereas mouse limb buds upregulate three myosin genes (Myh3, Myh7, Myh8) at E12.5, we detected only upregulation of Myh7 in bamboo shark fin buds. This is probably an example that our transcriptome data revealed an interesting divergence between sharks and mice.

Thus, our transcriptome data contain biologically meaningful information that allows us to discuss many aspects of differences and conserved processes between the shark fins and mouse limbs, and we carefully draw conclusions within the limitation of our data. We have added these discussions in the text and related data to Figure 1—figure supplement 8.

Another concern is the scaling (maximum TPM=1, minimum TPM=0). While I agree that this helps capturing the dynamics of gene expression, it does not reflect the magnitude of the change and could lead to misleading interpretations. I think this may happen with the interpretation of the Shh pathway

We apologize for the poor explanation for why we scaled TPMs. While we agree that the scaling loses information that can be extracted only from intact TPMs, it has a significant advantage when comparing between evolutionarily distant species besides capturing the dynamics of gene expression. Indeed, other past comparative transcriptome studies also scaled gene expression values in their own ways. Of those, z-score (or standardization) may be a relatively common strategy and better reflect the magnitude of changes in TPM values (e.g., Levin et al., 2016; Leiboff and Hake, 2019). However, we found that z-score has a disadvantage that it could misrepresent differences between species.

To validate our scaling method with an objective measure, we have examined the effect of intact TPM values and scaling methods by analyzing the expression dynamics of several housekeeping genes which exhibit a stable expression profile and are likely conserved in all vertebrates (these genes are listed in the BUSCO dataset; Simão et al., 2015) and that of heterochronically expressed genes. Namely, we assumed that the euclidean distance of expression values is close to zero if we only use housekeeping genes (left panel of Figure 1—figure supplement 6C), but it is larger when comparing heterochronic genes (right panel of Figure 1—figure supplement 6C). The result is that Max1 scored the highest ratio of the euclidean distance of the housekeeping genes to that of the heterochronic genes (Figure 1—figure supplement 6D), suggesting that Max1 is most sensitive to interspecific differences of dynamically regulated genes. Please see the main text, Figure 1—figure supplement 5,6, and section 1 in Supplementary file 7.

Regarding the SHH pathway shown in Figure 2 (now moved to Figure 3), we compared the temporal dynamics with intact TPM values, and did not find any misinterpretation caused by the scaling (please see Figure 3—figure supplement 1 and the main text). In addition, the *in situ* data for *Ptch1* and *Hoxd12* that we have added as a response to another comment are consistent with the RNA-seq data (Figure 3). Please note that comparing intact TPM values between species involves many technical uncertainties particularly when comparing distantly related species. For example, the difference in DNA sequences of transcripts (such as GC contents) between species probably affects the efficiency of library preparations and sequencing. TPM values are also very likely to be biased by the incompleteness of the reference transcriptome sequence that we used for the bamboo shark (e.g., some genes lack 3ʹUTR). Therefore, the fraction to the maximum TPM values (probably reflect the saturation point of genes) or the deviation to the average TPM values are more relevant biological indices than intact TPM values (please see section 1 in Supplementary file 7 for details).

1) Figure 2A: More information on the list of genes in each of the categories after gene-by-gene comparison of expression dynamics should be provided and discussed rather than only mentioning a couple of genes. The list should also be provided (i.e. excel). Particularly interesting is the inverse behavior of some genes in the "Heterochrony" group that are downregulated over time in the mouse limb bud being upregulated in the shark fin.

We apologize, the list of genes related to Figure 2A was originally shown in Supplementary file 3. However, as pointed by the reviewer 2, the table was not readily understandable. Therefore, we have improved the table to be readable and consistent with Figure 2A (Supplementary file 8).

As requested, we also sought genes that are downregulated over time in the mouse limb bud and being upregulated in the shark fin (Figure 2—figure supplement 1B for a heatmap visualization and Supplementary file 9 for the full list of genes). As a result, we identified 241 genes. Of those, Fgf8 is particularly interesting as it functions as a crucial growth signal from the apical ectodermal ridge (AER). On the other hand, in the zebrafish, *FGF24* and *Fgf16* are known to be indispensable signaling molecules of the apical fin fold (roughly equivalent to the AER). As shown in Figure 2—figure supplement 1C, *Fgf9, Fgf16* and *Fgf24* (known to be expressed in the AER/apical fin fold), instead of *Fgf8*, are expressed in the early stages of bamboo shark fin buds. Although we cannot infer the ancestral state of the expression pattern of these genes, the overlapping function of these genes may have allowed subfunctionalization. We have added these discussions in section 3 in Supplementary file 8.

2) Figure 2D-E- I think that some hybridizations for Shh and its main targets in the bamboo shark are required to sustain the conclusions from these two panels. It may be that the sustained expression in the mouse corresponds to later chondrogenic stages that have already started at E12.5 and the whole heterochrony responding to different time resolution as mentioned by the authors. I don't agree with the authors in that the expression dynamics of HoxA/D genes is similar in both species, at least for the 5' members.

As requested, we have added in situ hybridization data for *Ptch1* to visualize Shh pathway. In addition, we also examined *Hoxd12* expression pattern (a) to highlight the temporal difference of *Ptch1* expression and (b) to confirm the conserved dynamics of Hox genes. Overall, we concluded that these data are consistent with the transcriptome data. For some technical reasons, we failed to obtain in situ data for *Shh* on bamboo shark fin buds, even though we tried three different RNA probes (all resulted in non-specific expression pattern). However, we believe that *Ptch1* and *Hoxd12* are enough to address the comment raised here.

Please note that although we have prepared for a stable supply of bamboo shark embryos from an aquarium since 2015, the number of available embryos is still limited compared to other model species. Therefore, while we wanted to examine all the related genes, there is a fundamental limitation for it.

Regarding the HOX gene expression, the overall similarity of Hox gene expressions between the two species is supported by the measurement of euclidean distance as shown in Figure 4C. In addition, the *in situ* data for *Hoxd12* mentioned above also show a conserved expression dynamics, in which *Hoxd12* expression is initially restricted in the posterior part of fin/limb buds and extend anteriorly in later stages. Therefore, at least in some aspects, the transcriptional dynamics of Hox genes is comparable between these species.

That being said, we also agree that there are some differences in *Hox* gene expressions between these species. To address this, we have re-examined the 5ʹ members and described some differences that we think important. Namely, in the mouse limb bud, 5ʹ genes are known to exhibit a quantitative collinearity that the expression amount of *Hoxd13* is much higher than other neighbors, whose transcription levels decrease with distance from *Hoxd13*. On the other hand, in the bamboo shark fin bud, *Hoxd13* expression is lowest and there is no clear trend like quantitative collinearity (Figure 1—figure supplement 8; again, we emphasize that we cannot perform in situ for all of these genes due to the limited availability of bamboo shark embryos). Therefore, some degree of gene regulatory differences may exist between the two species. Accordingly, we have modified the manuscript to mention the difference of HOX gene expression (main text and section 2 in Supplementary file 7). Nevertheless, we think that our argument on the overall similarity of the spatio-temporal dynamics of HOX genes is well supported by the euclidean distance measure (Figure 4C) and the in situ data for *Hoxd12* (Figure 3C and D).

3) The consideration of ATAC sequences as active sequences should be softened as this is not always hold true. My interpretation is that most changes happen between 9.5 and 10.5, rather than conserved sequences being more active at E10.5

As suggested, we have softened the word “active” to either “open” or “accessible”.

Regarding the temporal dynamics of evolutionarily conserved open chromatin regions, we have analyzed how the number of conserved sequences differs between stage-specific open chromatin regions in two ways. First, we plotted the absolute count of accessible conserved sequences at each stage (Figure 5—figure supplement 2A and B). In these plots, accessible conserved sequences are clearly increased at E10.5 compared to other stages. Note that because there is a difference in total peak numbers between stages, the fraction of conserved sequences in peaks is also worth to account for. To do so, we showed the fraction of accessible conserved sequence counts in Figure 5D in the previous submission. Second, we also counted the number of conserved sequences in E10.5-specific peaks by using the output of clustering analysis indicated in Figure 5A. As a result, the E10.5-specific cluster (c7) contains the highest fraction of accessible conserved sequences (Figure 5—figure supplement 2C and D). In addition to these results, we have already shown in Figure 5B that many ATAC-seq peak signals are decreased and increased from E10.5 to E11.5. Therefore, conserved sequences are certainly more accessible at E10.5 than other stages.

Reviewer #2:This is an interesting study which aims to identify differences between fins and limbs. Performing transcriptomic comparison between pectoral fins from a non-model Chondricthyan species and forelimbs from mouse across a series of developmental stages, Onimaru et al. show that a noteworthy number of genes shows a heterochronic shift, alias a reverse temporal dynamic of expression between the two species. Moreover, they present an hourglass-shaped conservation of gene expression, but also of active regulatory regions in middle stages of development. Interestingly, in these stages they also detect more tissue- and stage-specific enhancers leading to the hypothesis that the middle developmental stages are evolutionary constrained by the increased regulatory complexity over pleiotropic genes.This work shows how comparing distant species constitutes a good approach to understand how morphological novelties occur or are constrained during evolution and hints towards some of the changes that might have occurred during fin-to-limb transition. The data that Onimaru et al. have produced are also a good resource for the scientific community and the overall work leads to many interesting follow-up questions. Due to all the above reasons, I support the publication of this article in eLife.However, the analyses presented are not always described as clearly or in-depth as desired and few observations are overstated. Therefore, I recommend the implementation of the comments below to strengthen the reliability, to enrich the content of the data presented and to prevent any confusion for the reader.1) In Figure 2A, using hierarchical clustering the authors show that there is a heterochronic shift in gene expression between mouse limbs and shark fins. However, this group consists of different subclusters which not all follow exactly the general trend that the authors describe in their results. Could the authors discuss on these genes that still show different temporal dynamics of expression between sharks and mouse, but do not show opposite -timewise- trend than in mouse?

We agree that the original cluster contained several subclusters that exhibit different temporal dynamics. To improve the clustering analysis, we used the UMAP method, which resulted in two major heterochronic groups (Heterochonic1 and 2 in Figure 2A). Of them, “Heterochronic2” mostly follows the previously identified trend (the c7 better fits with such trend than c6). “Heterochonic1” includes genes whose expression is highest during the late stages of mouse limb bud development but relatively stable throughout fin bud development. These genes are also interesting, and we examined the expression pattern for *Aldh1a2* as an example of genes in “Heterochonic 1”. A retinoic acid-generating enzyme, *Aldh1a2* is known to be expressed in the interdigital web and retinoic acid signaling regulates the interdigital cell death, which is very likely to be specific to the tetrapod limb. On the other hand, we found that in bamboo shark fin buds, *Aldh1a* transcripts are initially uniform and later restricted to the edge of fin buds (arrowheads in Figure2—figure supplement 1A). We have mentioned this in the Results.

In addition, as requested by the reviewer 1, we separately investigated genes that follow the inverse trend (downregulated over time in the mouse limb bud being upregulated in the shark fin; Figure 2—figure supplement 1B and Supplementary file 9). Of those, Fgf8 is particularly interesting as it plays a crucial role as a growth signal from the apical ectodermal ridge (AER). On the other hand, in the zebrafish, *FGF24* and *Fgf16* are known to be indispensable signaling molecules of the apical fin fold (roughly equivalent to the AER). As shown in Figure 2—figure supplement 1C, *Fgf9* (known to be expressed in mouse AER), instead of *Fgf8*, is expressed in the early stages of bamboo shark fin buds. Although we cannot infer the ancestral state of the expression pattern of these genes, the overlapping function of these genes may have allowed subfunctionalization. We have added these discussions in section 3 in Supplementary file 7.

2) The authors should provide individual tables for each cluster described in Figure 2A (fin-specific, limb-specific, stable, conserved/late, heterochronic) instead of the Supplementary file 3, which is quite confusing in its current form.

We apologize for the confusing table. We have examined how the table should be presented. If the table is split into the cluster annotations as requested, we think that there will be too many supplementary tables. Instead, we have added the cluster annotations to the original form of the table and the cluster numbers to the Figure 2A to be consistent.

3) In Figure 5, which are the GO terms associated to the genes for these clusters? What are the enriched motifs in cluster 8, largely specific to E9.5, and the GOs of the associated genes? The full list of motifs and associated genes for each cluster should be available. Moreover, is the conservation degree of the ATAC peaks different for each cluster?

We thank the reviewer for suggesting us an interesting analysis. We have performed GO analysis and the result was quite consistent with the clustering analysis. That is, while the constitutionally accessible peaks (C5, C6) are closely located to genes related to “cellular components”, the dynamically regulated peaks (C3, C4, C7, C8) are associated with GO terms, “developmental process”, “multicellular organism development”, and “anatomical structure morphogenesis”. Although there are subtle differences between C3, C4, C7 and C8, we did not detect any C7-specific GO terms. As requested, we have made a supplementary file that lists top 10 GO terms for every clusters (Supplementary file 12), and a supplementary file that contain full GO terms and all associated genes (Supplementary data) and mentioned this in the text.

As for the C8, the enriched motifs are mostly similar to those in the C7 (both consist of TAATT, ATTTAT and GACCTC), though the best matched transcription factors are different (VSX2 for C8 and LHX for C7). LHX and VSX motifs include TAATT sequence, but VSX1 and VSX2 are hardly expressed in mouse limb buds. As requested, we now showed the list of top 5 motifs of all clusters (Figure 6—figure supplement 2 and 3) and source data that include all motif list of all clusters (Supplementary data). In addition, for a better understanding of how motifs are dynamically changing over time, we have added plots of the average number of the motifs identified against mouse limb stages (Figure 6—figure supplement 5). These plots showed that COUP-TFII detected in C8 sharply decreases from E11.5. On the other hand, VSX2 (detected in C8) and LHX (detected in C7) follow a similar trend in which the number of the motif transiently increases at E10.5, suggesting that these motifs may represent a mostly same sequence feature. We have added this discussion in the text.

Regarding the conservation level of each cluster, E10.5-specific cluster (C7) is indeed enriched for conserved sequences as expected. We have added bar plots in Figure 5—figure supplement 2 and discussed in the text.

4) As far as the HOMER analysis is concerned in Figure 5, why the authors used the extended sequence length of 1400 bp to perform TF motif analysis? Also, did they perform the enrichment analysis with default HOMER options? If so, random genomic regions were used as a statistical background. Can these results be replicated when using a more biologically relevant background? For example, the peaks of Cluster5 are enriched in CTCF when compared to random regions of the DNA (the default HOMER approach), but are they also enriched in CTCF when compared to all the open chromatin regions that were detected during development?

We apologize that the HOMER analysis was poorly explained in the manuscript. First, we performed the motif analysis using +-50 bp from the center of a peak region by setting the “-size 100” option (the default of the program is “-size 200” , but we did not see any significant differences between -size 100 and 200). Therefore, although the input sequence length was 1400 bp, the regions that we analyzed are 100 bp. We have add the options that we used for running HOMER.

Second, we used random genomic regions as a background. We think that using random genomic regions is one way to fairly estimate enriched motifs, because some clusters share several enriched motifs (e.g., c5 and c6 are both enriched for the CTCF motif; please see Figure 6—figure supplement 2 and 3). However, as pointed by the reviewer, we also performed motif analyses with the other peak regions as a background (Figure 6—figure supplement 4), which consistently detected the CTCF motif.

5) Results first paragraph: the figure supplement 4, not 1, refers to the details of RNA-seq data.

We have corrected the figure number.

6) Have the authors used all three replicates in the transcriptomic analyses? We could assume that due to the last sentence referring to that in Figure 1—figure supplement 4, but it should be clearly stated.

We apologize for not mentioning this point. We used all three replicates, and means and standard errors of them were used for the all downstream analyses. We have corrected sentences to explicitly describe how the data was processed.

7) Could the authors explain why they used only one of the replicates for the ATAC-seq hierarchical clustering in Figure 5A and comment whether the ATAC-seq peaks tested were present in all 3 replicates?

One of the ways to integrate replicates in this analysis could be to extract peaks shared by all replicates. However, we faced two problems to do that. First, we used summits (an output of the MACS2 peak caller) to estimate the location of the center of peaks, but summits are rarely overlapped between replicates due to the narrow range of summits. Second, as shown in Figure 5B, because the quality of our ATAC-seq data is not uniform among replicates, data with relatively poor quality affects the analysis.

Now, to ensure that Figure 6A is reproducible, we showed data that repeatedly performed the same analysis with different combinations of replicates (Figure 6—figure supplement 1 and the Results). In addition, we have also added plots of the average number of the motifs identified using all the three replicates of ATAC-seq peaks at each stage (Figure 6—figure supplement 4), which support the clustering result.

[Editors’ note: what follows is the authors’ response to the second round of review.]

Revisions:Given the recent publication of Dr. Woltering (https://pubmed.ncbi.nlm.nih.gov/32875118/), the authors may want to comment on this paper in relation with the Shh, Hox expressions they report.

As requested, we have discussed Woltering et al., 2020 in relation to the present and previous studies including ours. While we agree with the authors that an anterior expansion of the expression domain of genes regulated by SHH may have contributed to the substantial anatomical changes during the fin-to-limb transition (as we have already proposed in Onimaru et al., 2015), we partly disagree with their argument. They proposed that the absence and presence of the phase of the anterior expansion of *Hoxd13* correlate with the metapterygial morphologies. However, this argument is valid ONLY within species that they picked up (cichlids, lungfish and tetrapods). Indeed, such anterior expansion of *Hoxd13* expression has been already observed in the fin buds of *Polyodon*, the little skate, and the small-spotted catshark (Davis MC et al. 2007, Freitas R et al. 2007, Nakamura T, et al., 2015). In addition, their *Hoxd13* expression data of lungfish fin buds is still needed to be confirmed, as there is a conflicting report by another group (Johanson et al., 2007). Therefore, while there is an overall consensus that an anterior expansion of gene expression domains may have contributed to the substantial anatomical changes during the fin-to-limb transition, further studies are necessary in order to understand the involvement of *Hoxd13* regulatory changes in the future.

We suggest that the authors indicate, at least in Materials and methods, their failure to detect Shh expression by ISH. Knowing this may be of help for other researchers.

We have mentioned on Shh ISH in the Materials and methods section.

Results paragraph three: Figure 1—figure supplement 7 instead of 8?

Thank you for pointing this out. We have corrected the figure number.

Introduction: the expression of Hoxa11 and Hoxa13 is actually not conserved in fin development, because the critical spatial exclusion of their expression domains is NOT seen in fins, even though a distal bias of Hoxa13 expression is shared. Please correct.

We have included the fact that the expression domains of Hoxa11 and Hoxa13 are overlapped in fin buds. We also note that the separation of Hoxa11/13 expression domain may not be a critical factor that differentiate limbs from fins, because axolotl limb buds also exhibit a similar overlapping expression of these genes (Woltering et al., 2019).

It is surprising to find 16,442 orthologs between shark and mouse, given that 1-1 orthologs among a sample of 10 eutherian species finds only <8,000 orthologs. A comment on this finding might be in order.

First of all, we would like to clarify that the number of genes that were uniquely orthologous to mouse ones is 13005. The figure of “16,442” represents the number of orthologs mapped to all vertebrate genes we analyzed. Although we are not sure which paper the reviewer referred to, based on a very recent study that infers orthology relations between human genes and genes from each of 43 other vertebrate genomes, the number of orthologous pairs for each genome comparison ranges from 9775 to 17027 with the average of around 14000 (Hao et al., 2020). Therefore, the number of the orthologs we detected is reasonable. We have mentioned this in the text.

Results paragraph three: a non-colinear relationship in Hoxd gene expression levels also applies to chicken wings, where Hoxd12 is higher expressed than all the others, but I am not sure that was ever published. This could point to a scenario where mouse limb development is not as paradigmatic as it often seems.

We thank the reviewer for making us realize this point. Indeed, we found a report that the expression level of *Hoxd12* is slightly higher (around 30–40%) than that of *Hoxd13* in the presumptive autopod region of chick fore- and hind-limb buds (Yakushiji-Kaminatsui et al., 2018). Since they did not perform replicates, the significance of this expression difference is not clear. Thus, at least we can say that Hoxd12 and Hoxd13 show a nearly same level of transcripts in chick limb buds. On the other hand, in bamboo shark fin buds at stage 31, the expression level of *Hoxd12* is 5.6 times as high as that of *Hoxd13* (p-value, 0.00712). This sharp contrast of Hoxd12/Hoxd13 expression amount may represent a substantial difference between shark fin buds and chick limb buds. Therefore, chick limbs and bamboo shark fins are probably not in the same situation. Although this limited taxon sampling does not give us a solid conclusion about the ancestral state, we appreciate that the mouse limb is not the paradigmatic case of the tetrapod limb. We have discussed the divergent controls of Hoxd genes.

Subsection “Comparison of SHH signaling pathways in limb and fin buds”: it is hard to see how Shh delayed onset can be supported without a rigid mapping of developmental stages between fin and limb development.

We agree that the original sentence was misleading. We have changed the words and added more explanation about Shh delay.

Discussion paragraph three: Please add reference to Piasecka et al., 2013.

We have added the paper and modified the Discussion accordingly.

Discussion paragraph four: We think it is important to be precise here. The correct statement is that mutations affecting this stage have more dramatic fitness consequences, rather than that it is less susceptible to mutation. What creates this impression is that the substitution rate is less not necessarily the mutation rate, as the authors note in the next sentence.

We apologize for the confusing discussion. This whole paragraph was not intended to discuss either the mid-stage conservation or fitness. Instead, we wanted to discuss the significance of regulatory sequences in terms of human diseases. We realized that this paragraph was quite off-topic and could mislead the audience. Therefore, we have deleted the whole paragraph.